# Robust Fine-Tuning from Non-Robust Pre-trained Models: Mitigating Suboptimal Transfer with Epsilon-Scheduling

**Jonas Ngnawé**[1,2]* **Maxime Heuillet**[1,2] **Sabyasachi Sahoo**[1,2] **Yann Pequignot**[1,2]

**Ola Ahmad**[3] **Audrey Durand**[1,2,5] **Frédéric Precioso**[4] **Christian Gagné**[1,2,5]

[1]Université Laval (IID)   [2]Mila   [3]CortAIx Labs, Thales Digital Solutions
[4]Université Côte d'Azur, CNRS, INRIA, I3S, Maasai   [5]Canada CIFAR AI Chair

## Abstract

Fine-tuning pretrained models is a standard and effective workflow in modern machine learning. However, robust fine-tuning (RFT), which aims to simultaneously achieve adaptation to a downstream task and robustness to adversarial examples, remains challenging. Despite the abundance of non-robust pretrained models in open-source repositories, their potential for RFT is less understood. We address this knowledge gap by systematically examining RFT from such non-robust models. Our experiments reveal that fine-tuning non-robust models with a robust objective, even under small perturbations, can lead to poor performance, a phenomenon that we dub *suboptimal transfer*. In challenging scenarios (eg, difficult tasks, high perturbation), the resulting performance can be so low that it may be considered a transfer failure. We find that fine-tuning using a robust objective impedes task adaptation at the beginning of training and eventually prevents optimal transfer. However, we propose a novel heuristic, *Epsilon-Scheduling*, a schedule over perturbation strength used during training that promotes optimal transfer. Additionally, we introduce *expected robustness*, a metric that captures performance across a range of perturbations, providing a more comprehensive evaluation of the accuracy-robustness trade-off for diverse models at test time. Extensive experiments on a wide range of configurations (six pretrained models and five datasets) show that *Epsilon-Scheduling* successfully prevents *suboptimal transfer* and consistently improves expected robustness.

## 1 Introduction

Fine-tuning pretrained models (backbones) on a downstream task is the standard workflow in machine learning, spanning natural language processing (Koroteev, 2021) and computer vision (Goldblum et al., 2023). This workflow offers clear benefits: (i) reusing a single foundation model across tasks (Devlin et al., 2019), (ii) faster convergence, (iii) better generalization than training from scratch (Yosinski et al., 2014), and (iv) reduced computation (Weiss et al., 2016), especially when labelled data is scarce (Pan & Yang, 2010).

However, in high-stakes applications, adversarial vulnerability remains a major concern (Biggio et al., 2013; Goodfellow et al., 2015). Adversarial Training (AT) (Madry et al., 2018) and its variants (Zhang et al., 2019; Wang et al., 2020; Ding et al., 2020; Shafahi et al., 2019a; Wong et al., 2020) are the most successful empirical defenses (Croce et al., 2021). Robust fine-tuning (RFT) is the integration of these methods in fine-tuning on downstream tasks (Shafahi et al., 2019b; Liu et al., 2023; Xu et al., 2024; Hua et al., 2024). Unlike standard fine-tuning, RFT must balance task adaptation with robustness – a trade-off that makes it considerably harder (Xu et al., 2024).

Most prior works assume access to robust backbones (Hua et al., 2024; Liu et al., 2023; Xu et al., 2024); however, in practice, nearly all widely used pretrained models from public repositories are

---

*Corresponding author: jonas.ngnawe.1@ulaval.ca

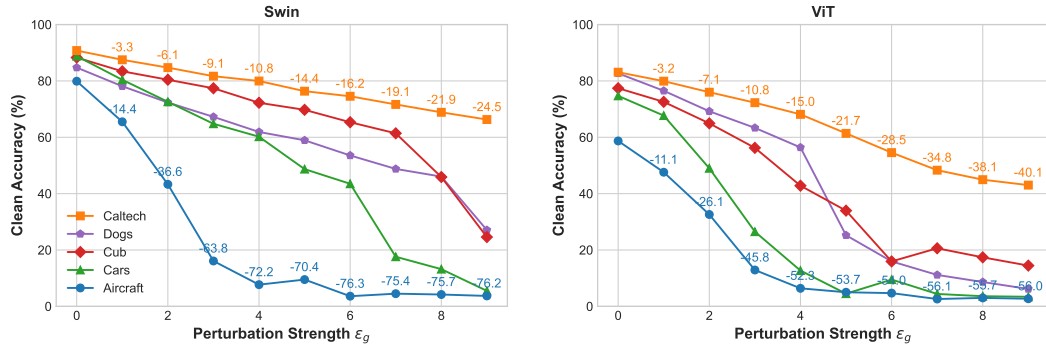

Figure 1: **RFT can lead to *suboptimal transfer* even when optimizing for small perturbation strenghts ($\varepsilon_g$).** The severity is highly model- and dataset-dependent.

non-robust (Wolf et al., 2020). Robust pretraining is costly and less common, and because pre-training pipelines typically prioritize broad general-purpose features, robustness is often treated as a property to be acquired downstream (Heuillet et al., 2025). This makes the development of RFT strategies for non-robust backbones not only consistent with current practice but also necessary to close the gap between research and deployment.

We adopt the standard RFT approach based on classical adversarial training (Madry et al., 2018), which optimizes a robust objective using adversarial examples generated at a target perturbation strength (commonly $4/255$ or $8/255$ in the $\ell_\infty$-norm). We apply this procedure to robustly fine-tune various backbones across multiple datasets and perturbation levels. Our experiments reveal that, even for small perturbation strengths, standard RFT often leads to **suboptimal transfer**: performance (clean accuracy) falls short of that achieved by standard fine-tuning (without perturbation) and is often too low to qualify it as a successful transfer. The severity of this effect depends on both the backbone and the downstream task (Figure 1).

When fine-tuning on a downstream task with a robust objective results in near-random performance, the benefits of using a pretrained model are diminished. This raises the question: do standard pre-trained models fail to offer a beneficial initialization for training a robust model? In this study, we explore the challenges associated with robust fine-tuning using standard pretrained models and pro-pose a novel approach to address these obstacles. In contrast to standard fine-tuning, where model adaptation to the downstream task occurs immediately, our study reveals that in robust fine-tuning, *task adaptation is delayed until later epochs*. This delay seems to eventually lead to *suboptimal transfer*, and we observe that the duration of the delay correlates negatively with transfer perfor-mance.

Based on our findings, we propose ***Epsilon-Scheduling***, a sched-ule over the perturbation strength during RFT to encourage op-timal transfer. This novel heuristic is a two-hinge linear sched-ule that begins with standard fine-tuning (zero perturbation) for early epochs and linearly increases to the target perturbation at final epochs (see Figure 2). This strategy effectively prevents *suboptimal transfer* and improves both accuracy and robustness.

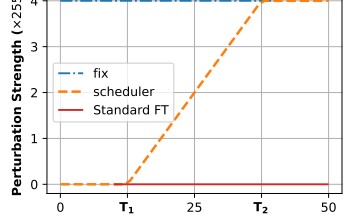

Figure 2: *Epsilon-Scheduling*

To better evaluate the fine-tuned models, we introduce a comple-mentary evaluation metric, dubbed ***expected robustness***. This proposed metric evaluates the expectation of the model accuracy across the full perturbation range from zero (clean accuracy) to the target robustness threshold. The *expected robustness* provides a concise, yet comprehensive measure of the accuracy–robustness trade-off, grounded in a practical threat model. We demonstrate that it offers valuable insights for model selection. Under this metric, *Epsilon-Scheduling*[1] consistently outperforms the standard robust-finetuning, even when worst-case robustness at the target threshold is similar or lower.

**Summary of Contributions**     Our main contributions are: **(i)** We show that robust fine-tuning from non-robust backbones often leads to *suboptimal transfer*, even at small perturbation strengths, where performance can fall significantly below standard fine-tuning. **(ii)** We find that robust finetuning re-sults in task adaptation delay compared to standard finetuning and that this delay strongly correlates with *suboptimal transfer*. **(iii)**We propose *Epsilon-Scheduling*, a two-hinge linear schedule to adjust

---

[1]Code available at: https://github.com/ngnawejonas/EpsilonScheduling

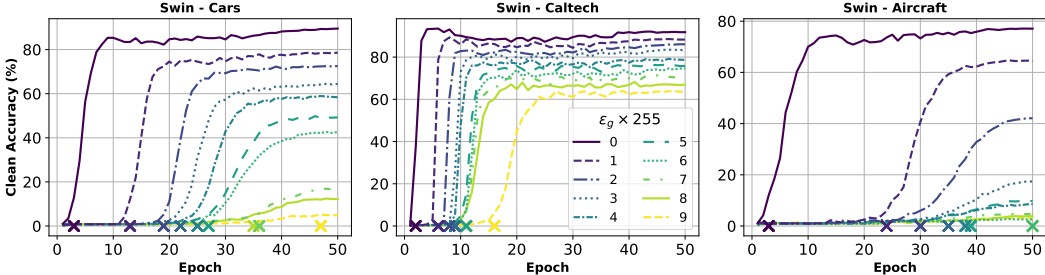

**Figure 3: RFT delays task adaptation.** Validation clean accuracy under standard fine-tuning ($\varepsilon_g = 0$) and RFT-`fix` with $\varepsilon_g \in [1/255, 9/255]$ on three datasets.The crosses indicate the onset of task adaptation (when validation accuracy exceeds 5%). Stronger perturbations cause longer delays and more severe suboptimal transfer. See Section 4 for analysis.

the training perturbation strength, which effectively mitigates the challenges associated with optimizing adversarial loss. **(iv)** We introduce *expected robustness*, a new evaluation metric capturing the full accuracy–robustness trade-off, and report performance using this metric for the first time. **(v)** Through extensive experiments, we show that *Epsilon-Scheduling* consistently prevents suboptimal transfer and improves expected robustness across both moderate ($4/255$) and high ($8/255$) perturbation regimes.

## 2 RELATED WORK

**Adversarial Robustness in Transfer Learning with Robust-FineTuning**   There are two main ways to achieve adversarial robustness in Transfer Learning: Robust Distillation (Goldblum et al., 2020; Dong et al., 2024) and Robust Fine-Tuning. Prior works on RFT focus on robust backbones (Liu et al., 2023; Xu et al., 2024; Hua et al., 2024). TWINS (Liu et al., 2023) employs two networks with shared parameters to separately track pretraining and downstream batch statistics. However, Liu et al. (2023) do not apply it to non-robust backbones claiming that "robust pre-training is indispensable to downstream robustness". AutoLoRA (Xu et al., 2024) disentangles natural and adversarial objectives using a LoRA branch for the former and a robust pretrained extractor for the latter, though it relies on TRADES loss (Zhang et al., 2019), which is harder to scale than standard adversarial training (Madry et al., 2018). Xu et al. (2024) show that robust pretrainning is necessary for AutoLoRA and register the worst performance for the non-robust pretraining. RoLi (Hua et al., 2024) preserves robustness by initializing the classifier head via adversarial linear probing before performing RFT. (Hua et al., 2024) explicitly argue that robust pretraining is a prerequisite for RoLi by showing that linear probing with a robust objective on a non-robust backbone fails dramatically and therefore does not provide a good initialization. In summary, all these approaches assume robust pretrained features. In contrast, we are the first to propose an RFT method targeting non-robust backbones.

**Tuning Perturbation Strength in Adversarial Training**   Adapting the perturbation strength during training has been explored in various forms. Early work used a linear ramp-up in Interval Bound Propagation (Gowal et al., 2018). Ding et al. (2020) connected margin maximization to minimal adversarial loss, motivating adaptive, sample-specific perturbation strengths, though such instance-wise selection (Balaji et al., 2019) is computationally costly. They also proposed PGDLS (PGD with Linear Scaling), which linearly increases perturbation strength but shows gains only at large perturbations ($24/255$). Other strategies include sampling the perturbation strength from a Beta distribution (Chamon & Ribeiro, 2020) and curriculum schemes that gradually increase the number of attack steps (Cai et al., 2018). Pang et al. (2021) reports that linear warmup provides limited gains in ResNets, whereas Debenedetti et al. (2023) finds that it improves both clean and robust accuracy in vision transformers. Unlike prior works, which apply to adversarial training from scratch, our study is on transfer learning. Our formulation generalizes linear warmup, and we show that the benefits consistently hold across tasks and architectures, including ResNets, using the new *expected robustness* metric.

**Adversarial Defense Evaluation**   Standard evaluation (Croce et al., 2021) compares clean and robust accuracy at a target perturbation strength under strong attacks or ensembles (Carlini & Wagner, 2017; Madry et al., 2018; Croce & Hein, 2020b; Cinà et al., 2025), yet it obfuscates what happens at intermediate perturbation strengths. A related recommendation is to check that accuracy decreases with stronger perturbations (Carlini et al., 2019), but this test is unquantified and serves only as an

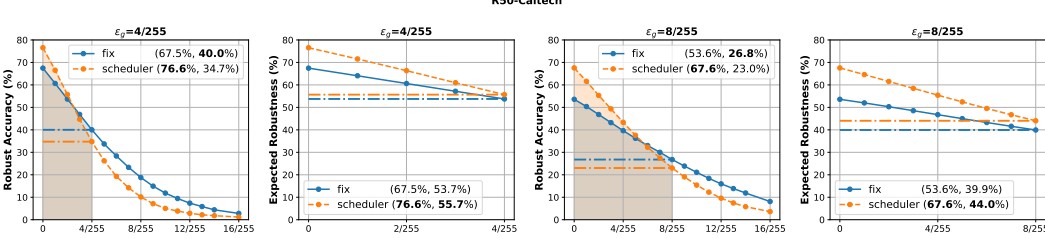

Figure 4: **The expected robustness metric offers a valuable perspective for model selection.** The larger the area under the curve (shaded area), the higher the expected robustness. The values in the legend indicate the clean accuracy and the evaluation at $\varepsilon_g$.

informal validation (Debenedetti et al., 2023). In contrast, our notion of *expected robustness* formalizes this decrease and interpolates between clean and worst-case performance at a specific target perturbation strength. Another class of metrics interpolates between worst-case and average-case robustness (Rice et al., 2021; Li et al., 2021), the latter defined against random or natural perturbations (Hendrycks & Dietterich, 2019; Han et al., 2024). However, this approach does not capture the trade-off between clean and worst-case performance.

## 3  BACKGROUND

**Supervised Fine-Tuning**   Consider a classification task to map instances $x$ of a $d$-dimensional input space $\mathcal{X} \subset \mathbb{R}^d$ to corresponding labels $y$ in the set $\mathcal{Y} = \{1, 2, ..., K\}$. Unlike training from scratch to learn a classifier $f_\theta : \mathcal{X} \mapsto \mathcal{Y}$ from randomly initialized parameters $\theta$ on a training dataset drawn iid from a data distribution $\mathcal{D}$ on $\mathcal{X} \times \mathcal{Y}$, supervised fine-tuning uses a pretrained feature extractor (backbone) $h_{\theta_1} : \mathcal{X} \mapsto \mathcal{Z}$ that maps inputs to a representation space $\mathcal{Z}$ and a randomly initialized classifier head $c_{\theta_2} : \mathcal{Z} \mapsto \mathcal{Y}$ such that $f_{\{\theta_1, \theta_2\}} = c_{\theta_2} \circ h_{\theta_1}$. This work focuses on full fine-tuning where both $\theta_1$ and $\theta_2$ are trainable parameters. The performance of the fine-tuned model is measured by its accuracy, i.e., the probability that a prediction is correct for an instance drawn from $\mathcal{D}$. We will refer to this as clean accuracy (or transfer accuracy).

**Adversarial Training (AT)**   Given a target evaluation threshold for perturbation strength $\varepsilon_{goal}$ ($\varepsilon_g$ for convenience) in $\ell_p$-norm ($\|x\|_p = \left( \sum_i x_i \right)^{1/p}$, $p > 0$), adversarial training aims to train a classifier $f$ such that it maximizes the robust accuracy $\mathrm{Acc}_{\varepsilon_g}(f)$. The robust accuracy is the probability that a prediction remains correct for any input $x$ under a perturbation $\delta$ of maximum norm $\varepsilon_g$. Classical adversarial training (Madry et al., 2018) minimizes the adversarial risk at a perturbation threshold $\varepsilon_g$ as a surrogate objective for $\mathrm{Acc}_{\varepsilon_g}(f)$:

$$R_{\varepsilon_g}(f) = \mathbb{E}_{(x,y) \sim \mathcal{D}} \left( \max_{\|\delta\|_p < \varepsilon_g} L_{\mathrm{CE}}(f(x + \delta), y) \right) \tag{1}$$

where $L_{\mathrm{CE}}$ is the cross-entropy loss. In practice, the empirical counterpart of the risk $R_\varepsilon(f)$ is minimized, and adversarial perturbations $\delta$ are generated using a few iterations of Projected Gradient Descent (PGD) under an $\ell_p$-norm constraint.

In this work, we refer to **robust fine-tuning** (RFT) as supervised fine-tuning with classical adversarial training. The standard practice in RFT for achieving robustness at a target evaluation threshold $\varepsilon_g$ is to directly minimize the empirical risk at $\varepsilon_g$ throughout the fine-tuning process (Madry et al., 2018; Hua et al., 2024). In this setup, the perturbation strength used to generate adversarial examples remains fixed at $\varepsilon_g$ across all fine-tuning epochs. We refer to this baseline strategy as RFT-`fix` (or `fix`).

## 4  CHARACTERIZING SUBOPTIMAL TRANSFERS IN ROBUST FINE-TUNING

When we perform robust fine-tuning with excessively strong perturbations $\varepsilon_g$ that severely corrupt the inputs, clean accuracy is expected to reach chance level, as the training samples will become unrecognisable, drifting away from the task data distribution $\mathcal{D}$ (Carlini et al., 2019). Yet, a question remains: how much clean accuracy degrades as $\varepsilon_g$ increases given a pretrained model and a dataset? To investigate this question, we conduct an experiment with two non-robust backbones: `SWIN` (Liu et al., 2021) and `ViT` (Dosovitskiy et al., 2021), across five datasets: `Cub` (Wah et al., 2011), `Dogs`

| Model | Dataset Setting | Aircraft Clean | Adv. | E. adv. | Caltech Clean | Adv. | E. adv. | Cars Clean | Adv. | E. adv. | Cub Clean | Adv. | E. adv. | Dogs Clean | Adv. | E. adv. |
|---|---|---|---|---|---|---|---|---|---|---|---|---|---|---|---|---|
| ViT | fix | 6.40 | 2.80 | 4.48 | 68.14 | 41.64 | 55.07 | 12.70 | 4.90 | 8.20 | 42.82 | 15.12 | 27.79 | 56.40 | **19.97** | 36.93 |
|  | sched | **58.60** | **13.20** | **34.95** | **78.73** | **41.69** | **60.71** | **73.40** | **19.10** | **46.71** | **73.40** | **23.63** | **48.09** | **70.69** | 15.69 | **41.62** |
| SWIN | fix | 7.70 | 4.80 | 6.11 | 79.97 | **57.16** | 69.19 | 60.20 | 29.70 | 44.74 | 72.25 | **41.87** | 57.55 | 61.89 | **26.89** | 44.17 |
|  | sched | **73.80** | **32.00** | **53.75** | **85.43** | 56.39 | **72.04** | **84.70** | **43.20** | **66.41** | **82.29** | 41.61 | **63.82** | **72.70** | 24.32 | **48.50** |
| CNX | fix | 7.60 | 4.50 | 5.86 | 83.27 | **61.54** | 73.08 | 69.60 | 43.20 | 57.52 | 76.34 | **47.08** | 62.59 | 68.90 | **31.61** | 50.61 |
|  | sched | **78.40** | **38.00** | **59.40** | **89.41** | 61.45 | **77.23** | **88.90** | **57.70** | **75.85** | **85.17** | 44.99 | **67.30** | **78.39** | 26.31 | **53.19** |
| R50 | fix | 8.40 | 2.90 | 4.56 | 67.47 | **40.02** | 53.74 | 4.20 | 2.90 | 3.49 | 49.19 | 19.35 | 33.58 | 57.05 | **19.80** | 37.73 |
|  | sched | **53.10** | **11.10** | **29.40** | **76.55** | 34.74 | **55.67** | **70.00** | **19.30** | **43.44** | **70.06** | **19.59** | **43.62** | **69.11** | 15.94 | **41.11** |
| ClipViT | fix | 5.00 | 3.30 | 4.16 | 31.91 | 15.49 | 23.00 | 4.90 | 3.00 | 3.74 | 13.95 | 3.64 | 7.97 | 7.89 | 3.29 | 5.39 |
|  | sched | **69.80** | **33.90** | **52.79** | **74.83** | **46.64** | **60.99** | **86.70** | **58.60** | **75.01** | **74.35** | **35.67** | **55.54** | **63.17** | **20.87** | **41.05** |
| ClipCNX | fix | 3.10 | 2.50 | 2.82 | 61.76 | 42.13 | 51.54 | 2.80 | 1.60 | 2.23 | 28.89 | 14.33 | 20.92 | 23.90 | 11.33 | 17.14 |
|  | sched | **81.70** | **50.70** | **67.88** | **81.19** | **52.68** | **67.71** | **90.90** | **74.10** | **84.33** | **79.06** | **42.11** | **61.45** | **70.85** | **25.85** | **48.19** |

Table 1: **At moderate perturbation regime** ($4/255$), ***Epsilon-Scheduling***, **mitigates** *suboptimal transfers* **and consistently improves expected robustness.** See Table 2 for $\varepsilon_g = 8/255$

(Khosla et al., 2011), `Caltech` (Griffin et al., 2007)), `Cars` (Krause et al., 2013)), and `Aircraft` (Maji et al., 2013)). See Appendix A for details on backbones and their pretraining.

***Suboptimal Transfer***   We apply RFT-`fix` with target perturbation strengths $\varepsilon_g$ in $[1/255, 9/255]$ and compare performance (clean accuracy) to standard fine-tuning ($\varepsilon_g = 0$). While low clean accuracies for RFT-`fix` from non-robust backbones have been reported on `Caltech` at $\varepsilon_g = 8/255$ (Liu et al., 2023; Hua et al., 2024), our experiments reveal a broader picture. Figure 1 shows distinct trends in clean accuracy degradation as $\varepsilon_g$ increases. Even for tiny perturbations ($\varepsilon_g = 1/255$), accuracy can drop by up to $14\%$ compared to the standard finetuning with $\epsilon_g = 0$. At the commonly used threshold $\varepsilon_g = 4/255$, the minimum drop is $10\%$. In extreme cases, RFT-`fix` fails to adapt effectively, with clean accuracy falling below $5\%$, making it practically unusable. We refer to this phenomenon as ***suboptimal transfer***, where RFT-`fix` yields a clean accuracy substantially lower than standard fine-tuning, at times to the point of ineffectiveness or failure.

**Insights on Task Difficulty and Backbone Selection**   The severity of *suboptimal transfer* depends on both the backbone and the task, with the task being the more differentiating factor (see Figure Figure 1). Easier tasks (e.g., `Caltech`, wide variety of objects) exhibit smaller performance drops than more challenging ones (e.g., `Aircraft`, highly similar categories). As expected, the choice of `SWIN` as backbone is better than `ViT` (Hua et al., 2024). However, the notion of "task difficulty" is also model-dependent: (`Dogs` < `Cub` for `SWIN` but `Dogs` > `Cub` for `ViT`). This demonstrates that task difficulty is a property that emerges from the interaction between model and dataset (Zilly et al., 2019; Jankowski et al., 2025), rather than by data features alone (Cao et al., 2024). Our experiments further suggest that with respect to adversarial robustness, the perturbation strength may also play a role: `Dogs` is easier than `Cub` for perturbations below $\varepsilon_g = 4/255$, but becomes comparable or more difficult at higher strengths.

**Robust Fine-Tuning Delays Task Adaptation**   Figure 3 shows that in standard fine-tuning ($\varepsilon_g = 0$), the model adapts to the downstream task almost immediately – validation accuracy rises from the first epoch – since there is no robustness constraint that competes with task adaptation. With $\varepsilon_g > 0$, RFT-`fix` distorts task-relevant features, which prevents early adaptation and delays the onset of task adaptation. For example, at $\varepsilon_g = 4/255$, adaptation begins around epoch 10 for `Caltech`, epoch 25 for `Cars`, and after epoch 30 for `Aircraft`. If we precisely measure the delay time as the epoch from which the validation accuracy starts being above $5\%$, the correlation between the delay times and the severity of the *suboptimal transfer* is very high (above $90\%$, see Appendix B.1 for details). The delay shortens the effective adaptation period, likely leading to *suboptimal transfer* with increased severity at higher perturbation thresholds. To the best of our knowledge, the delayed onset of task adaptation in robust fine-tuning has not been previously reported, which is an interesting insight that could open up new opportunities for improvement.

## 5 Epsilon-Scheduling and Expected Robustness

The analysis in section 4 demonstrates that a robust objective at high perturbation strengths is detrimental in RFT from a non-robust pretrained model, eventually leading to *suboptimal transfer*, with performance subpar of what is expected from the pretraining advantage. Based on this finding, we propose *Epsilon-Scheduling*, a simple schedule over the perturbation strength to mitigate this effect.

| Model | Dataset Setting | Aircraft Clean | Adv. | E. adv. | Caltech Clean | Adv. | E. adv. | Cars Clean | Adv. | E. adv. | Cub Clean | Adv. | E. adv. | Dogs Clean | Adv. | E. adv. |
|---|---|---|---|---|---|---|---|---|---|---|---|---|---|---|---|---|
| ViT | fix | 3.00 | 2.00 | 2.50 | 44.95 | 19.52 | 31.43 | 3.60 | 2.00 | 2.74 | 17.40 | 2.80 | 8.56 | 8.64 | 2.88 | 5.35 |
|  | sched | **57.00** | **6.70** | **27.72** | **72.86** | **26.89** | **49.28** | **68.10** | **9.00** | **35.18** | **64.74** | **9.79** | **33.93** | **56.86** | **5.79** | **25.81** |
| SWIN | fix | 4.20 | 2.70 | 3.47 | 68.87 | 38.10 | 53.40 | 13.20 | 5.60 | 8.66 | 45.89 | 13.60 | 28.56 | 46.05 | **11.08** | 26.69 |
|  | sched | **69.20** | **22.40** | **45.12** | **80.27** | **38.67** | **60.26** | **78.00** | **23.50** | **53.57** | **74.80** | **21.07** | **47.34** | **60.49** | 8.73 | **31.14** |
| CNX | fix | 1.60 | 1.50 | 1.48 | 59.85 | 33.95 | 46.34 | 5.30 | 2.60 | 3.98 | 5.02 | 2.28 | 3.56 | 27.33 | 7.73 | 16.28 |
|  | sched | **75.00** | **28.80** | **50.90** | **84.99** | **41.82** | **64.92** | **85.60** | **35.90** | **65.04** | **80.69** | **24.28** | **53.07** | **68.94** | **9.78** | **36.51** |
| R50 | fix | 1.30 | 0.90 | 0.74 | 53.59 | 26.78 | 39.93 | 1.50 | 1.20 | 1.34 | 30.89 | 8.27 | 17.84 | 27.14 | **6.95** | 15.61 |
|  | sched | **42.80** | **5.30** | **20.38** | **67.56** | 23.01 | **44.03** | **57.10** | **8.50** | **29.56** | **59.49** | **8.68** | **29.95** | **50.89** | 6.92 | **25.26** |
| ClipViT | fix | 3.60 | 2.20 | 3.05 | 23.02 | 7.29 | 14.52 | 3.00 | 2.50 | 2.73 | 11.11 | 2.30 | 5.73 | 2.20 | 1.38 | 1.77 |
|  | sched | **65.80** | **25.40** | **44.84** | **70.68** | **33.70** | **51.67** | **84.70** | **38.60** | **64.47** | **67.64** | **18.05** | **41.79** | **54.28** | **8.94** | **27.78** |
| ClipCNX | fix | 1.80 | 1.30 | 1.62 | 51.94 | 28.37 | 39.44 | 1.30 | 1.10 | 1.25 | 6.37 | 2.30 | 4.05 | 8.36 | 3.97 | 5.98 |
|  | sched | **79.20** | **34.50** | **59.09** | **76.53** | **37.20** | **56.83** | **90.00** | **55.20** | **77.14** | **73.58** | **22.75** | **47.77** | **62.67** | **11.36** | **33.85** |

Table 2: **At high perturbation regime ($8/255$), RFT-`fix` mostly fails and *Epsilon-Scheduling* preserves performance.** See Table 1 for $\varepsilon_g = 4/255$.

***Epsilon-Scheduling***  In contrast with RFT-`fix`, we perform RFT for target perturbation strength $\varepsilon_g$ by minimizing an empirical counterpart of $R_\varepsilon(f)$ where $\varepsilon$ follows a schedule during the fine-tuning, given for each epoch $t$ as a proportion $\alpha(t) \geq 0$ of $\varepsilon_g$: $\varepsilon(t) = \alpha(t)\varepsilon_g$. We propose a two-hinge linear scheduler illustrated in Figure 2 and defined by:

$$\alpha(t) = \begin{cases} 0 & : t < T_1 \\ \frac{t-T_1}{T_2-T_1} & : T_1 \leq t < T_2 \\ 1 & : t \geq T_2. \end{cases}$$

This strategy begins by training over $T_1$ epochs without robustness ($\varepsilon = 0$), then linearly increases from $\varepsilon = 0$ to $\varepsilon = \varepsilon_g$ over $T_2 - T_1$ epochs, to finally remain constant ($\varepsilon = \varepsilon_g$) from epoch $T_2$. Note that this generalizes the linear warmups mentioned in prior work (Pang et al., 2021; Debenedetti et al., 2023) for $T_1 = 0, T_2 \neq T_1$, while falling back to RFT-`fix` for $T_1 = T_2 = 0$.

From a transfer learning perspective, this strategy can be viewed as follows: *begin with task adaptation, then gradually shift to the robust objective, and conclude by minimizing the target robust objective*. Here, $T_1$ denotes the adaptation phase, i.e., the time for the model to reach good clean accuracy, while $T_2$ controls the steepness of the transition from 0 to $\varepsilon_g$. In the following, we refer to this strategy as RFT-`scheduler` (or simply `scheduler`). Considering that stronger perturbations make task adaptation harder (i.e., cause adaptation delays), RFT-`scheduler` can be viewed as a curriculum learning strategy that first exposes the model to easier examples before gradually introducing harder ones (Cai et al., 2018; Pang et al., 2021; Debenedetti et al., 2023).

**Expected Robustness Evaluation**  While RFT targets low adversarial risk $R_{\varepsilon_g}(f)$, models are usually evaluated both for clean accuracy $\text{Acc}_0(f)$ and robust accuracy $\text{Acc}_{\varepsilon_g}(f)$. We propose to extend this classical evaluation to take into account intermediary perturbation strengths within the range $[0, \varepsilon_g]$ and define the *expected robustness* metric as the expectation under uniform distribution $U$ of the accuracy over $[0, \varepsilon_g]$:

$$\text{Acc}_{[0,\varepsilon_g]}(f) := \mathbb{E}_{\varepsilon \sim U[0,\varepsilon_g]}\big[\text{Acc}_\varepsilon(f)\big] = \frac{1}{\varepsilon_g}\int_0^{\varepsilon_g} \text{Acc}_\varepsilon(f)\, d\varepsilon = \frac{1}{\varepsilon_g}\text{AUC}_{\varepsilon_g}(f),$$

where $\text{AUC}_{\varepsilon_g}(f)$ represents the area under the accuracy curve from 0 to $\varepsilon_g$ (Figure 4, panels 2 and 4). This can be estimated using the trapezoidal rule to numerically approximate the integral. See Appendix A for additional details. When comparing two models with similar accuracies, particularly in the presence of a clean–robust trade-off, the distinct patterns observed at intermediate perturbation strengths (Figure 4, panels 1 and 3) can inform model selection, which *expected robustness* summarizes quantitatively.

The expected robustness metric also evaluates performance under a more realistic threat model where inputs may or may not be altered. Clean accuracy corresponds to the idealized case where inputs are never perturbed, while robust accuracy at $\varepsilon_g$ assumes that all inputs are perturbed with a budget of $\varepsilon_g$. In contrast, expected robustness estimates the accuracy when perturbations of size up to $\varepsilon_g$ are applied at random (here, uniformly). While our choice of a uniform distribution serves as a good option from a use-case-agnostic perspective, the distribution of perturbations can be tailored to align with the relevant threat model for a specific application. With the uniform distribution, each adversarial strength is weighted equally, however, for example, with a Dirac distribution centered at $0$ ($\epsilon_g$), it falls back to clean accuracy (worst case robust accuracy).

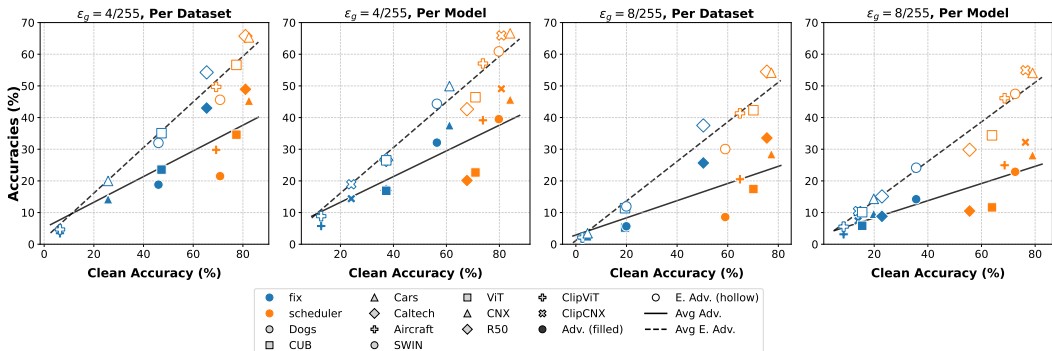

Figure 5: *Epsilon-Scheduling* **mitigates** *suboptimal transfers* **and consistently improves** *expected robustness* **even when robust accuracy is equivalent.** Aggregated results from Table 1 and Table 2.

## 6 EXPERIMENTAL EVALUATION

We provide an overview of the experimental setup–including backbones, datasets, parameters $T_1$ and $T_2$ for *Epsilon-Scheduling*, and training and evaluation procedures. Additional details for reference and reproducibility are provided in Appendix A.

**Backbones:** We perform experiments using six non-robust backbones, selected to cover two prominent architecture families (attention-based and convolutional-based) and two pretraining paradigms (supervised and multi-modal). Transformers: *Swin-Base* (CNX, (Liu et al., 2021) and *ViT-Base* (Dosovitskiy et al., 2021); convolutional architectures: *ConvNext-Base* (Liu et al., 2022) and *ResNet-50* (He et al., 2016); CLIP models (Radford et al., 2021): *CLIP-ViT* and *CLIP-ConvNext*.

**Downstream Datasets:** We evaluate fine-tuning performance on five low-data downstream tasks: bird species classification on **CUB-200-2011** (Wah et al. (2011), 200 classes), dog breed classification on **Stanford Dogs** (Khosla et al. (2011), 120 classes), object recognition on **Caltech256** (Griffin et al. (2007), 257 classes), car model classification on **Stanford Cars** (Krause et al. (2013), 196 classes), and aircraft variant classification on **FGVC-Aircraft** (Maji et al. (2013), 100 classes).

**Choice of $T_1$ and $T_2$ for *Epsilon-Scheduling*:** To obtain values of $T_1$ and $T_2$ that are general enough for most cases, we use measurements from the most severe instance of *suboptimal transfer* in Section 4 (SWIN-Aircraft). We define the adaptation phase as the epoch when validation accuracy reaches 90% of its final value, which occurs at epoch 11. Accordingly, we set $T_1 = 12$, corresponding to about 25% of the total training epochs, sufficient for the model to reach high clean accuracy. Since the average task-adaptation delay in RFT-fix is observed around epoch 37, we set $T_2 = 37$, i.e., roughly 75% of the total epochs, so that the model is trained with perturbation strengths smaller than $\varepsilon_g$ during the delay period.

**Training and Evaluation** We follow a similar setup described in Hua et al. (2024). We train for 50 epochs and generate adversarial examples using APGD (Croce & Hein, 2020b) (instead of PGD) with cross-entropy loss as in prior work (Singh et al., 2023; Heuillet et al., 2025), which removes the need for manual tuning. The number of APGD steps is 7 for training and 10 for evaluation (Hua et al., 2024). Results are reported for the models at the end of training because overfitting of the adversarial accuracy is negligible (Figure 6). The evaluation is conducted in the $\ell_\infty$-norm, which is the most widely studied norm in the literature (Croce et al., 2021; Ngnawe et al., 2024), using two standard evaluation thresholds: $\varepsilon_g = 4/255$ (moderate perturbation) and $\varepsilon_g = 8/255$ (high perturbation). For each model, we report the clean accuracy (**clean**), APGD accuracy (**adv.**), and the expected APGD accuracy (E. adv.) over the interval $[0, \varepsilon_g]$. We provide in Appendix B.2 a few additional results for SWIN, with the more expensive evaluation AutoAttack (Croce & Hein, 2020b), a diverse ensemble of four attacks containing untargeted APGD-CE, targeted APGD-DLR, targeted FAB (Croce & Hein, 2020a), and black-box Square Attack (Andriushchenko et al., 2020).

### 6.1 EPSILON-SCHEDULING PERFORMANCE IN RFT

Results are reported in Table 1 for the moderate perturbation regime ($\varepsilon_g = 4/255$), in Table 2 for the high perturbation regime ($\varepsilon_g = 8/255$), and in Figure 5 for the aggregated analysis.

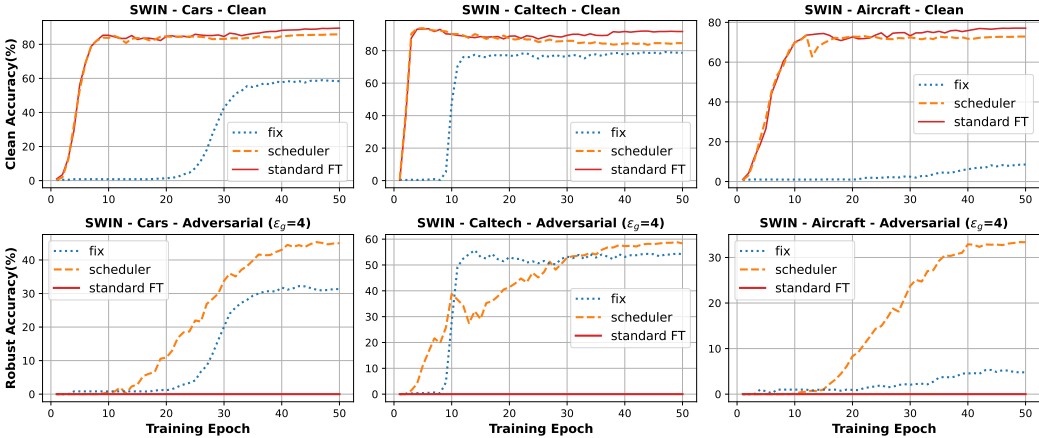

Figure 6: *Epsilon-Scheduling* **preserves task adaptation while improving robustness (⁴/₂₅₅).**

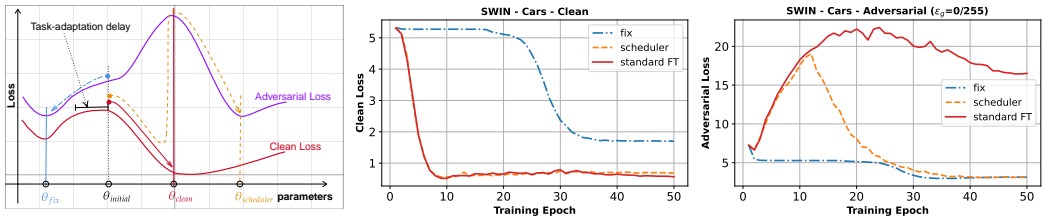

Figure 7: *Epsilon-Scheduling* **discovers a different local optimum.** *Left*: Illustrative example of the difference between RFT-`fix` and RFT-`scheduler`. *Center and right*: Evolution of validation loss (Clean and Adversarial) for the `SWIN` backbone on the `Cars` dataset with $\varepsilon_g = {}^4/_{255}$.

**Moderate Perturbation Regime** ($\varepsilon_g = {}^4/_{255}$)  Table 1 shows that while RFT-`fix` often fails to transfer with low clean accuracy, RFT-`scheduler` achieves high clean accuracy for most models and maintains a decent adversarial accuracy. While RFT-`fix` sometimes achieves better adversarial accuracy (in 9 out of 30 configurations), our scheduling strategy consistently yields higher clean and expected accuracy. These results show that even at moderate perturbations (⁴/₂₅₅), *Epsilon-Scheduling* prevents the steep degradation incurred by RFT-`fix`, allowing models to retain strong clean performance while achieving improved or similar adversarial accuracy at non-trivial levels.

**High Perturbation Regime** ($\varepsilon_g = {}^8/_{255}$)  At stronger perturbations, performance naturally declines, as shown in Table 2. RFT-`fix` almost always fails to transfer, yielding very low accuracies. In contrast, RFT-`scheduler` consistently improves clean accuracy and achieves higher expected robustness in all 30 configurations. For adversarial accuracy alone, the scheduler outperforms in 28 out of 30 cases.

Overall, as shown in the aggregated results (Figure 5), *Epsilon-Scheduling* consistently improves expected robustness through significant gains in clean accuracy, even when a robustness–accuracy trade-off exists or when robustness is similar across datasets and backbones. This contrasts with linear warmups in adversarial training from scratch, which benefit vision transformers but not residual networks (Pang et al., 2021; Debenedetti et al., 2023).

## 6.2 RESULTS ANALYSIS

In order to further analyze *Epsilon-Scheduling*, we consider three datasets representing different levels of task difficulty, as determined by the severity of suboptimal transfer in Section 4: `Aircraft` (high), `Cars` (medium), and `Caltech` (low).

***Epsilon-Scheduling* promotes task adaptation while improving robustness**  Figure 6 shows the validation accuracy during training. Standard fine-tuning quickly reaches high clean accuracy without robustness, whereas RFT-`fix` improves robustness but degrades clean accuracy. In contrast, RFT-`scheduler`, achieves a high clean accuracy during the first stage and once perturbation strengths start passing above zero, robust accuracy rises while clean accuracy remains surprisingly high and stable.

| Model | Setting | $\varepsilon_g = {}^4/_{255}$ | | | | | | | | | $\varepsilon_g = {}^8/_{255}$ | | | | | | | | |
| | | Aircraft | | | Caltech | | | Cars | | | Aircraft | | | Caltech | | | Cars | | |
| | | Clean | Adv. | E. Adv. | Clean | Adv. | E. Adv. | Clean | Adv. | E. Adv. | Clean | Adv. | E. Adv. | Clean | Adv. | E. Adv. | Clean | Adv. | E. Adv. |
| RobViT | fix | 63.00 | **43.30** | **53.29** | 78.67 | 57.73 | 68.59 | 72.90 | **49.30** | **62.23** | 51.60 | **25.10** | 37.91 | 73.16 | **41.89** | 57.48 | 62.10 | **24.40** | 43.16 |
| | sched | **63.50** | 34.80 | 49.14 | **82.22** | **57.93** | **70.86** | **78.00** | 43.40 | 62.05 | **65.10** | 23.00 | **42.88** | **79.57** | 41.12 | **60.97** | **78.60** | 24.30 | **52.39** |
| RobSWIN | fix | 74.00 | **56.50** | **66.19** | 82.64 | **61.12** | 72.59 | 83.10 | **60.10** | 72.42 | 71.60 | **43.30** | 57.84 | 77.39 | **45.87** | 61.95 | 67.00 | 31.90 | 50.62 |
| | sched | **77.20** | 49.90 | 64.99 | **84.88** | 58.71 | **73.11** | **86.60** | 54.20 | **73.15** | **74.90** | 38.00 | 57.44 | **81.86** | 43.43 | **63.88** | **84.20** | **39.10** | **63.88** |
| RobCNX | fix | 78.30 | **62.00** | **70.74** | 85.20 | **67.11** | 77.03 | 86.10 | **68.40** | **78.69** | 74.70 | **48.00** | **62.22** | 79.77 | **49.87** | 65.50 | 80.30 | **50.00** | 67.51 |
| | sched | **80.20** | 49.70 | 65.59 | **87.92** | 66.89 | **78.53** | **88.40** | 63.90 | 77.99 | **78.00** | 36.50 | 57.07 | **85.74** | 48.93 | **69.35** | **87.60** | 45.10 | **70.38** |
| RobR50 | fix | 63.10 | **41.80** | **52.14** | 71.25 | **48.83** | 60.29 | 72.10 | **45.80** | 59.61 | 60.30 | **32.50** | 45.62 | 65.25 | **36.60** | 50.19 | 59.60 | **24.30** | 41.89 |
| | sched | **66.50** | 36.20 | 51.14 | **75.49** | 47.60 | **61.79** | **78.10** | 44.20 | **62.29** | **66.90** | 24.70 | 44.26 | **71.90** | 34.08 | **52.35** | **76.80** | 22.90 | **50.37** |

Table 3: *Epsilon-Scheduling* **on robust backbones.** The `scheduler` (sched) improves clean accuracy at the cost of a decrease in robustness achieved in `fix`, but overall, the expected robustness is still improved.

**Insight on the optimization process with *Epsilon-Scheduling*** From an optimization standpoint, RFT-`scheduler` seems to converge to a distinct local minimum of the adversarial loss compared to the one achieved by RFT-`fix`, as illustrated in Figure 7 (left). The local minimum attained by RFT-`scheduler` is notably characterized by a lower value of the clean loss, whilst reaching a comparable value of the adversarial loss around epoch $T_2 = 37$. Xu et al. (2024) found that the gradient of clean loss and the adversarial loss can point in opposite directions. Our experiments appear to confirm that this indeed happens when initializing at a non-robust pretrained model. We observe that standard fine-tuning effectively minimizes the clean loss (Figure 7, center), but this comes at the expense of increasing the adversarial loss (Figure 7, right). In contrast, during the first 20 epochs, RFT-`fix` appears to struggle to reduce the adversarial loss while the clean loss remains nearly equal to the adversarial loss. However, the optimization trajectory of RFT-`scheduler` initially aligns with that of standard fine-tuning, resulting in a low clean loss value. Subsequently, RFT-`scheduler` effectively reduces the adversarial loss while maintaining a minimal degradation in clean loss. This allows RFT-`scheduler` to achieve a balance that appears difficult for RFT-`fix`.

**Effect *Epsilon-scheduling* on robust backbones** Table 3 shows that robust backbones are indeed more resilient to large perturbations under RFT-`fix` than their non-robust counterparts, reducing the need for *Epsilon-Scheduling*. Nevertheless, RFT-`scheduler` still consistently boosts clean accuracy relative to RFT-`fix`, although at the cost of reduced robustness. On the easy task (`Caltech`), the trade-off is in favour of the `scheduler`. A key takeaway is that *Epsilon-Scheduling* substantially reduces the large clean-accuracy gap previously observed between RFT from non-robust backbones and their robust equivalents (Liu et al., 2023; Hua et al., 2024), even if robustness at target $\epsilon_g$ is not fully matched.

## 6.3 ABLATION AND SENSITIVITY ANALYSIS

We summarize the effect of the hyperparameters $T_1$ and $T_2$ of *Epsilon-Scheduling* in Appendix C.1 as follows. *(i)* $T_2$ have the most significant influence with the control of steepness $(1/(T_2-T_1), T_1 \neq T_2)$. When $T_2$ is close to $T_1$, clean accuracy decreases, whereas robust accuracy increases; this eventually leads to suboptimal transfer. This is in line with the motivation for linear warmup in Debenedetti et al. (2023), although they do not study this effect. *(ii)* Increasing $T_1$ increases clean accuracy, up to a threshold beyond which further increasing $T_1$ has no apparent effect.

**Special cases** *Only delaying the robust objective* without follwing with gradual linear increase, i.e., a schedule that switches directly from 0 to $\varepsilon_g$ ($T_1 > 0, T_1 = T_2$), is unstable: validation accuracy drops sharply to its initial value and does not recover during training unless $T_1$ is small enough. *Linear warmups* ($T_1 = 0, T_2 > 0$) without the delay still improve over `fix`, provided $T_2$ is sufficiently large to ensure low steepness, thus having only very small perturbations early in training to avoid distorting features. The *end-to-end linear* schedule ($T_1 = 0, T_2 = 50$) comes close to the performance of the `scheduler`, though the latter remains superior.

**Targeting directly the expected robustness** A possible strategy to directly minimize the expected robustness risk $\left(\mathbb{E}_{\varepsilon \sim U[0,\varepsilon_g]} R_\varepsilon(f)\right)$ is via Monte Carlo estimation with a single sample, which is equivalent to training with an $\varepsilon$ randomly drawn from $U[0, \varepsilon_g]$ at each epoch. Results in Appendix C.2 show that the random uniform strategy (`uniform`) often results in *suboptimal transfer*, except on relatively easy datasets such as Caltech. This behaviour is normal: the expected perturbation strength is $\varepsilon_g/2$, making it likely that high perturbation levels appear early in training, thereby impeding effective transfer.

| Model | Setting | $\epsilon = 4/255$ | | | | | | | | | $\epsilon = 8/255$ | | | | | | | | |
|---|---|---|---|---|---|---|---|---|---|---|---|---|---|---|---|---|---|---|---|
| | | Aircraft | | | Caltech | | | Cars | | | Aircraft | | | Caltech | | | Cars | | |
| | | Clean | Adv. | E.Adv. | Clean | Adv. | E.Adv. | Clean | Adv. | E.Adv. | Clean | Adv. | E.Adv. | Clean | Adv. | E.Adv. | Clean | Adv. | E.Adv. |
| SWIN | fix | 7.70 | 4.80 | 6.11 | 79.97 | 57.16 | 69.19 | 60.20 | 29.70 | 44.74 | 4.20 | 2.70 | 3.47 | 68.87 | 38.10 | 53.40 | 13.20 | 5.60 | 8.66 |
| | sched | 73.80 | 32.00 | 53.75 | 85.43 | 56.39 | 72.04 | 84.70 | 43.20 | 66.41 | 69.20 | 22.40 | 45.12 | 80.27 | 38.67 | 60.26 | 78.00 | 23.50 | 53.57 |
| | auto | 73.30 | 29.40 | 52.96 | 85.63 | 54.18 | 71.29 | 84.20 | 38.40 | 64.30 | 68.40 | 18.60 | 42.69 | 81.71 | 35.82 | 59.92 | 79.20 | 18.70 | 51.43 |
| CNX | fix | 7.60 | 4.50 | 5.86 | 83.27 | 61.54 | 73.08 | 69.60 | 43.20 | 57.52 | 1.60 | 1.50 | 1.48 | 59.85 | 33.95 | 46.34 | 5.30 | 2.60 | 3.98 |
| | sched | 78.40 | 38.00 | 59.40 | 89.41 | 61.45 | 77.23 | 88.90 | 57.70 | 75.85 | 75.00 | 28.80 | 50.90 | 84.99 | 41.82 | 64.92 | 85.60 | 35.90 | 65.04 |
| | auto | 79.10 | 31.60 | 56.61 | 90.14 | 58.30 | 76.26 | 89.00 | 50.30 | 72.56 | 76.20 | 23.60 | 49.65 | 86.48 | 38.39 | 64.35 | 86.20 | 29.90 | 63.49 |

Table 4: **Results on an automated scheduler derived from our analysis** for SWIN and ConvNext (CNX) on Aircraft, Caltech, and Cars, at $\epsilon = 4/255$ (left block) and $\epsilon = 8/255$ (right block).

**Automated Scheduler** Based on our analysis, we can derive a simple *automatic epsilon-scheduler (auto)* driven by the validation accuracy. The procedure starts with $\epsilon = 0$ and then initiates a linear increase from $T_1$ to the end of training, where $T_1$ is automatically selected as the point at which the validation accuracy converges. Convergence is detected by monitoring the change in validation accuracy with patience of 5 epochs and a tolerance of 2%. Table 4 presents the results obtained with this automatic scheduler, which show that although it has less expected robustness compared to RFT-`scheduler`, it effectively mitigates suboptimal transfer and provides strong performance across tasks.

## 7 CONCLUSION

We present the phenomenon of *suboptimal transfer* in robust fine-tuning from non-robust backbones and its connection with delayed task adaptation. To address this, we propose *Epsilon-Scheduling*, a heuristic perturbation schedule over perturbation strength, and demonstrate that it effectively mitigates this phenomenon, using commonly used metrics as well as the introduced *expected robustness*. Our findings underscore the practical potential of scheduling in robust transfer learning and motivate further exploration of fine-tuning strategies from non-robust pretrained backbones.

**Limitations and Future Work.** Although *Epsilon-Scheduling* yields significant improvements, robustness can still be limited even when clean accuracy is high, highlighting the potential for future research to further enhance performance. This work opens doors to exploring other scheduling strategies, either heuristic, theoretically motivated, or learning-based. Extending the analysis to other vision tasks, such as detection or segmentation, applying the framework to parameter-efficient methods like LoRA, and investigating whether similar dynamics occur in other modalities, such as natural language processing, remain open questions. Studying these cases may require special considerations such as task-specific losses or hyperparameters.

From a theoretical perspective, although we offer an explanation based on the discrepancy between the clean and robust loss landscapes in the vicinity of the pretrained model, a deeper understanding of robust fine-tuning in this setting remains an open challenge. Our findings point to several important open problems: **(i)** What mechanisms underlie suboptimal transfer: is delayed task adaptation the only cause of suboptimal transfer or are there other factors? **(ii)** Can we find other approaches to mitigate delayed task adaptation different from Epsilon-Scheduling? **(iii)** What mathematical theory can account for suboptimal transfer or delayed task adaptation? **(iv)** If robust pretraining is not indispensable, what specific properties (if any) in pretraining really matter for downstream robustness and allow effective robust fine-tuning?

Pursuing these directions promises to unlock more effective strategies for robust fine-tuning and yield more substantial progress towards achieving robustness in downstream tasks.

ACKNOWLEDGEMENTS

This work is supported by the DEEL Project CRDPJ 537462-18 funded by the Natural Sciences and Engineering Research Council of Canada (NSERC) and the Consortium for Research and Innovation in Aerospace in Québec (CRIAQ), together with its industrial partners Thales Canada inc, Bell Textron Canada Limited, CAE inc and Bombardier inc.[2]

---

[2]https://deel.quebec

## REPRODUCIBILITY STATEMENT

Our study is designed to be fully reproducible. All backbones and datasets are publicly available, with details and references provided in Section 6 and Appendix A, where we also cite the prior work underlying our design choices. Details on the estimation of expected robustness are given in Appendix A.

We provide an anonymized GitHub repository containing the implementation, the results of the hyperparameter optimization, all the data used to generate the paper's figures and tables, and a script to reproduce them. The repository also includes step-by-step instructions for downloading datasets and pretrained models, creating Python environments, and launching experiments.

Finally, details on compute resources and expected run times are reported in Appendix A.

Link to public github repository: https://github.com/ngnawejonas/EpsilonScheduling

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

| Shorthand (Configuration Name) | HuggingFace ID | References |
|---|---|---|
| **ViT** (vit_b,sup,in1k) | timm/vit_base_patch16_224.augreg_in1k | Steiner et al. (2022) |
| **SWIN** (swin_b,sup,in22k-in1k) | timm/swin_base_patch4_window7_224.ms_in22k_ft_in1k | Liu et al. (2021) |
| **CNX** (convnext_b,sup,in22k-in1k) | timm/convnext_base.fb_in22k_ft_in1k | Liu et al. (2022) |
| **ClipViT** (vit_b,clip,laion2b) | timm/vit_base_patch16_clip_224.laion2b | Ilharco et al. (2021) |
| **ClipCNX** (convnext_b,clip,laion2b) | laion/CLIP-convnext_base_w-laion2B-s13B-b82K | Schuhmann (2022) |
| **R50** (resnet50,sup,in1k) | timm/resnet50.a1_in1k | Wightman et al. (2021) |
| **RobCNX** (robust_convnext_b,sup,in1k) | | Liu et al. (2025) |
| **RobSWIN** (robust_swin_b,sup,in22k-in1k) | | Liu et al. (2025) |
| **RobR50** (robust_resnet50,sup,in1k) | | Liu et al. (2025) |
| **RobViT** (robust_vit_b,sup,in1k) | | Liu et al. (2025) |

Table 5: Pretrained non-robust and robust models used with HuggingFace IDs and references. The model name indicates the architecture ({vit, swin, convnext, resnet50}), the training type (sup: supervised, clip: multimodal), and the dataset: **in1k** = ImageNet-1k, **in22k** = ImageNet-22k, **in22k-in1k** = pretrained on ImageNet-22k then fine-tuned on ImageNet-1k, **laion2b** = LAION-2B.

## APPENDIX

## A  EXPERIMENTAL SETUP DETAILS

**Pretrained Models**   The non-robust backbones come from *timm* (PyTorch Image Models) and are publicly available on HuggingFace. The robust models are publicly released by ARES and can be accessed at github.com/thu-ml/ares/. A summary of all models used in this work is provided in Table 5.

**Training Splits and Data Augmentations**   We use train-val-test split from Hua et al. (2024) for `Caltech`, `Cub`, `Stanford Dogs`; and from Heuillet et al. (2025) for `Aircraft` and `Stanford Cars`. Training augmentations consist of standard preprocessing methods commonly used for ImageNet and high-resolution images (Marcel & Rodriguez, 2016): random horizontal flips ($p = 0.5$), color jitter (brightness, contrast, and saturation set to $0.25$), and random rotations. As done in Robustbench (Croce et al., 2021), images are resized to 224x224 with pixel values in the range $[0, 1]$, and data normalization and standardization are directly integrated into the model.

**Hyperparameters Optimization**   We use the AdamW optimizer with a cosine learning rate scheduler that includes a warmup period. We select the learning rate and weight decay via hyperparameter optimization (HPO) based on clean accuracy. HPO is performed only for the `fix` setting, and the resulting hyperparameters are reused for the `scheduler` setting to ensure a fair comparison. We search learning rate and weight decay values in the range $10^{-5}$ to $10^{-1}$, using the ASHAS scheduler, a variant of Hyperband Li et al. (2018).. The exploration budget is 30 minutes for all configurations. HPO results are available in the code repository.

**Additional Evaluation Details**   The expected robustness is estimated by using the trapezoidal rule with evaluations made with steps $1/255$, so for example with $\varepsilon_g = 4/255$:

$$\text{AUC}_{4/255}(f) = \frac{1}{4} \sum_{i=0}^{3} \frac{\text{Acc}_{\frac{i}{255}}(f) + \text{Acc}_{\frac{i+1}{255}}(f)}{2}.$$

**Compute Resources**   Experiments were conducted using a 4xNVIDIA H100 GPU with 80GB of Memory. The duration for a single case of robust fine-tuning ranges from approximately 15 minutes to one hour in distributed mode. An evaluation of robust accuracy for $\varepsilon_g$ from 0 to 16 can run in 5 minutes or less with APGD. The same evaluations with AutoAttack require a minimum of 4 hours; the most expensive models can go up to 24 hours or more.

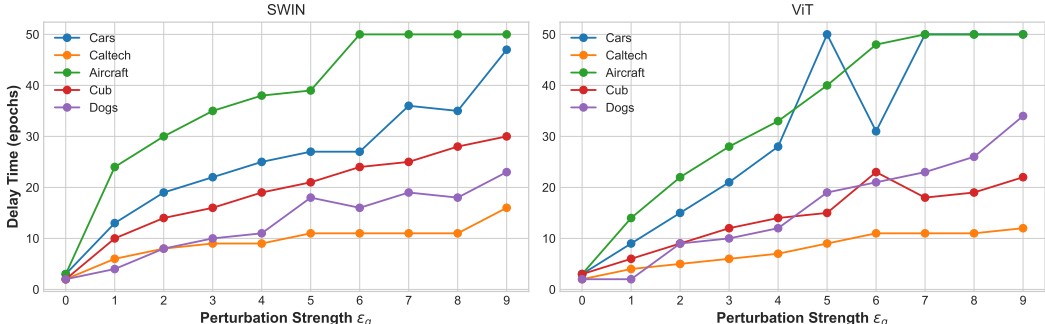

Figure 8: **Delay times increases with perturbation strength**. We take the delay time here as the epoch from which the validation accuracy starts being above $5\%$. In some cases, the model never goes beyond this threshold until the end of training at 50 epochs. See Section 4

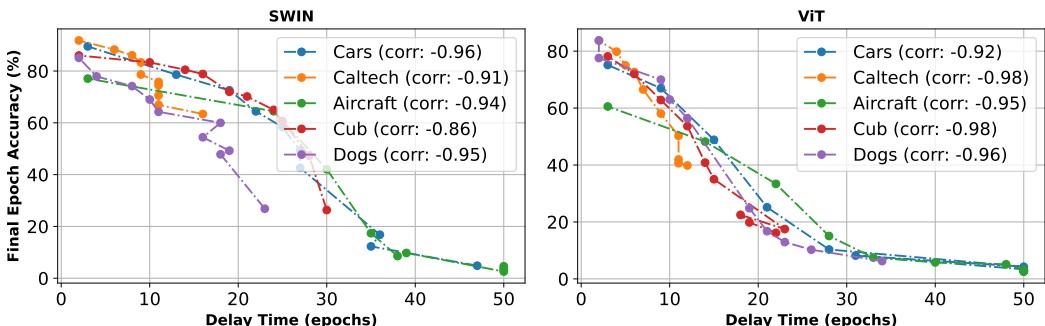

Figure 9: **Delay times strongly correlates with *suboptimal transfer* performance**. The final validation accuracy is lower because task adaptation starts at later epochs. See Section 4

# B    ADDITIONAL RESULTS

## B.1    TASK ADAPTATION DELAYS

We report detailed results on the increase in task adaptation delay time with growing perturbation strength (Figure 8), as well as the correlation between delay times and the severity of suboptimal transfer (Figure 9).

## B.2    AUTOATTACK RESULTS

AutoAttack (Croce & Hein, 2020b) is a stronger and more diverse attack on the models, but is more expensive. We evaluate a few cases (`SWIN` on $\{$`Cars`, `Aircraft`$\}$ x $\{4/255, 8/255\}$). Results can be found in Table 6 and Figure 10. Although it takes substantially more time, the results are close to evaluations with APGD.

# C    ABLATION AND SENSITIVITY ANALYSIS

## C.1    ABLATION AND SENSITIVITY ANALYSIS

To evaluate the influence of $T_1$ and $T_2$ on the performance of *Epsilon-Scheduling*, we consider multiple configurations, illustrated in Figure 11 (moderate perturbation, $4/255$) and Figure 12 (high perturbation, $8/255$). These figures illustrate the evolution of validation losses and accuracies during training, along with test set evaluations, showcasing the distinct trends. The corresponding numerical results on the test set are reported in Table 7.

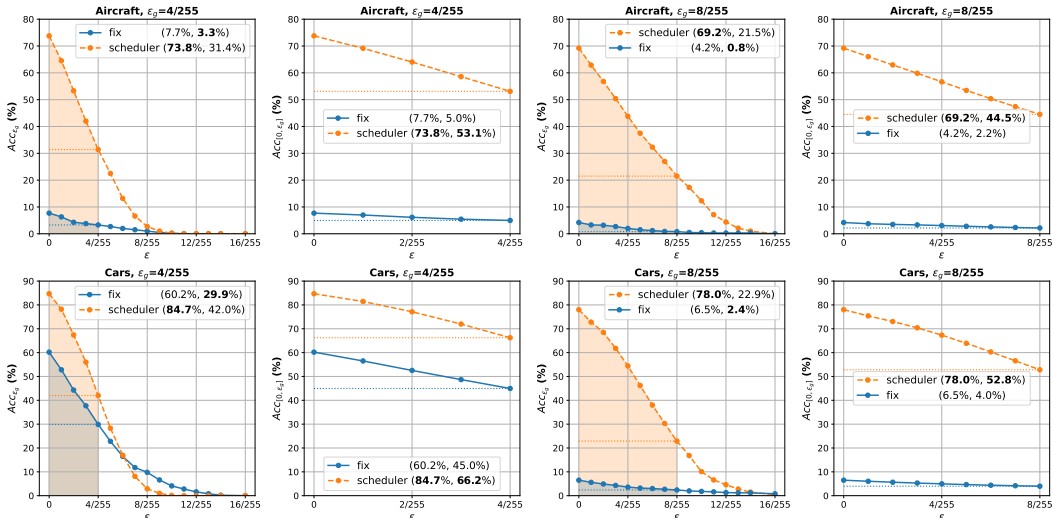

Figure 10: Evaluation with AutoAttack. Numerical values are in Table 6.

| $\epsilon$ | Attack | Setting | Aircraft | | | Cars | | |
|---|---|---|---|---|---|---|---|---|
| | | | Clean Acc | Adv. | E. Adv. | Clean Acc | Adv. | E. Adv. |
| $4/255$ | APGD | fix | 7.70 | 4.80 | 6.11 | 60.20 | 31.9 | 45.89 |
| | | sched | 73.80 | 32.00 | 53.75 | 84.70 | 43.20 | 66.41 |
| | AutoAttack | fix | 7.70 | 3.30 | 4.97 | 60.20 | 29.90 | 44.96 |
| | | sched | 73.80 | 31.40 | 53.10 | 84.70 | 42.00 | 66.24 |
| $8/255$ | APGD | fix | 4.20 | 2.70 | 3.47 | 6.50 | 3.2 | 4.49 |
| | | sched | 69.20 | 22.40 | 45.12 | 78.00 | 23.50 | 53.57 |
| | AutoAttack | fix | 4.20 | 0.80 | 2.16 | 6.50 | 2.40 | 3.97 |
| | | sched | 69.20 | 21.50 | 44.51 | 78.00 | 22.90 | 52.81 |

Table 6: AutoAttack results

## C.2 DIRECT MINIMIZATION FOR EXPECTED ROBUSTNESS

Since *Epsilon-Scheduling* consistently improves expected robustness, we compare with a direct minimization of the expected robustness risk. The results in Table 8 show *Epsilon-Scheduling* is still superior, and the uniform strategy often leads to suboptimal transfer due to early sampling of high perturbations.

## D STATISTICAL SIGNIFICANCE

We report paired $t$-test statistics comparing **RFT-Fix** and **RFT-Scheduler** at $\epsilon = 4/255$ and $\epsilon = 8/255$ in Table 9. These tests assess whether performance differences between the two strategies are statistically significant across downstream tasks. A paired $t$-test measures whether the mean performance difference between two methods is reliably non-zero; small $p$-values indicate that the observed differences are unlikely to occur by chance.

We also report the averages for each metric per model (Table 10) and per dataset (Table 11).

## E ADDITIONAL RESULTS ON IMAGENETTE

We provide additional results for ImageNette in Table 12

| T1 | T2 | $\varepsilon_g = 4/255$ | | | $\varepsilon_g = 8/255$ | | |
|----|----|-------|------|--------|-------|------|--------|
|    |    | Clean | Adv. | E. Adv. | Clean | Adv. | E. Adv. |
| 0  | 0  | 60.20 | 31.90 | 45.89 | 6.50  | 3.20  | 4.49  |
|    | 12 | 67.70 | 40.20 | 54.19 | 36.20 | 10.50 | 21.47 |
|    | 30 | 78.40 | 44.70 | 63.41 | 63.30 | 21.80 | 43.34 |
|    | 50 | 82.30 | 40.30 | 63.95 | 75.40 | 19.90 | 49.33 |
| 5  | 5  | 64.20 | 29.50 | 47.26 | 4.20  | 2.40  | 3.26  |
|    | 12 | 78.40 | 48.20 | 64.95 | 68.00 | 24.20 | 47.06 |
|    | 25 | 81.10 | 48.60 | 66.64 | 74.80 | 26.00 | 52.54 |
|    | 50 | 84.50 | 35.90 | 63.32 | 80.30 | 18.00 | 51.58 |
| 12 | 12 | 1.90  | 1.40  | 1.74  | 1.30  | 1.30  | 1.28  |
|    | 30 | 83.00 | 47.30 | 67.09 | 78.10 | 26.60 | 55.06 |
|    | 37 (*) | 84.70 | 43.20 | 66.41 | 78.00 | 23.50 | 53.57 |
|    | 50 | 84.80 | 35.80 | 63.25 | 81.10 | 16.60 | 51.79 |
| 25 | 25 | 0.80  | 0.80  | 0.80  | 0.80  | 0.80  | 0.80  |
|    | 37 | 84.00 | 39.50 | 64.61 | 78.40 | 21.00 | 51.51 |
|    | 50 | 84.30 | 24.80 | 57.46 | 81.00 | 12.10 | 46.94 |

Table 7: Effect of hyperparameters $T_1$ and $T_2$. The training dynamics can be found in Figure 11 for $\varepsilon_g = 4/255$ and Figure 12 for $\varepsilon_g = 8/255$. (*) RFT-`scheduler` reported in main text ($T_1 = 12, T_2 = 37$).

| Model | Setting | $\varepsilon_g = 4/255$ | | | | | | | | | $\varepsilon_g = 8/255$ | | | | | | | | |
|-------|---------|-------|------|--------|-------|------|--------|-------|------|--------|-------|------|--------|-------|------|--------|-------|------|--------|
|       |         | Aircraft | | | Caltech | | | Cars | | | Aircraft | | | Caltech | | | Cars | | |
|       |         | Clean | Adv. | E. Adv. | Clean | Adv. | E. Adv. | Clean | Adv. | E. Adv. | Clean | Adv. | E. Adv. | Clean | Adv. | E. Adv. | Clean | Adv. | E. Adv. |
| SWIN | fix | 7.70 | 4.80 | 6.11 | 79.97 | **57.16** | 69.19 | 60.20 | 29.70 | 44.74 | 4.20 | 2.70 | 3.47 | 68.87 | 38.10 | 53.40 | 13.20 | 5.60 | 8.66 |
|      | uniform | 30.10 | 7.90 | 18.57 | 83.45 | 55.49 | 70.27 | 70.90 | 35.30 | 54.68 | 7.80 | 2.00 | 4.85 | 76.74 | 37.31 | 57.34 | 53.70 | 11.30 | 30.16 |
|      | sched | **73.80** | **32.00** | **53.75** | **85.43** | 56.39 | **72.04** | **84.70** | **43.20** | **66.41** | **69.20** | **22.40** | **45.12** | **80.27** | **38.67** | **60.26** | **78.00** | **23.50** | **53.57** |
| R50 | fix | 8.40 | 2.90 | 4.56 | 67.47 | **40.02** | 53.74 | 4.20 | 2.90 | 3.49 | 1.30 | 0.90 | 0.74 | 53.59 | **26.78** | 39.93 | 1.50 | 1.20 | 1.34 |
|     | uniform | 41.50 | 8.80 | 22.04 | 74.95 | 34.68 | 54.49 | 43.10 | 8.90 | 23.28 | 27.40 | 3.70 | 12.26 | 67.55 | 22.03 | 43.44 | 6.20 | 2.10 | 3.36 |
|     | sched | **53.10** | **11.10** | **29.40** | **76.55** | 34.74 | **55.67** | **70.00** | **19.30** | **43.44** | **42.80** | **5.30** | **20.38** | 67.56 | 23.01 | **44.03** | **57.10** | **8.50** | **29.56** |
| ClipCNX | fix | 3.10 | 2.50 | 2.82 | 61.76 | 42.13 | 51.54 | 2.80 | 1.60 | 2.23 | 1.80 | 1.30 | 1.62 | 51.94 | 28.37 | 39.44 | 1.30 | 1.10 | 1.25 |
|         | uniform | 7.10 | 4.30 | 5.78 | 72.25 | 47.07 | 59.66 | 8.10 | 3.90 | 5.92 | 3.10 | 2.20 | 2.68 | 61.78 | 30.35 | 45.09 | 3.50 | 1.30 | 2.28 |
|         | sched | **81.70** | **50.70** | **67.88** | **81.19** | **52.68** | **67.71** | **90.90** | **74.00** | **84.33** | **79.20** | **34.50** | **59.09** | **76.53** | **37.20** | **56.83** | **90.00** | **55.20** | **77.14** |

Table 8: *Epsilon-Scheduling* **still has better expected robustness than a direct optimization for the expected robustness risk.** In fact the approximation with `uniform` can still lead to *suboptimal transfer*.

| Metric | n_pairs | $\epsilon = 4/255$ | | $\epsilon = 8/255$ | |
|--------|---------|--------|---------|--------|---------|
|        |         | $t$-stat | $p$-value | $t$-stat | $p$-value |
| Clean Acc | 30 | 7.823294 | $1.255470 \times 10^{-8}$ | 12.387491 | $4.170254 \times 10^{-13}$ |
| Adv. | 30 | 4.348780 | $1.540867 \times 10^{-4}$ | 5.447550 | $7.317049 \times 10^{-6}$ |
| E. Adv. | 30 | 6.595568 | $3.153155 \times 10^{-7}$ | 9.270810 | $3.572919 \times 10^{-10}$ |

Table 9: $t$-test statistics between RFT-fix and RFT-scheduler for two perturbation magnitudes.

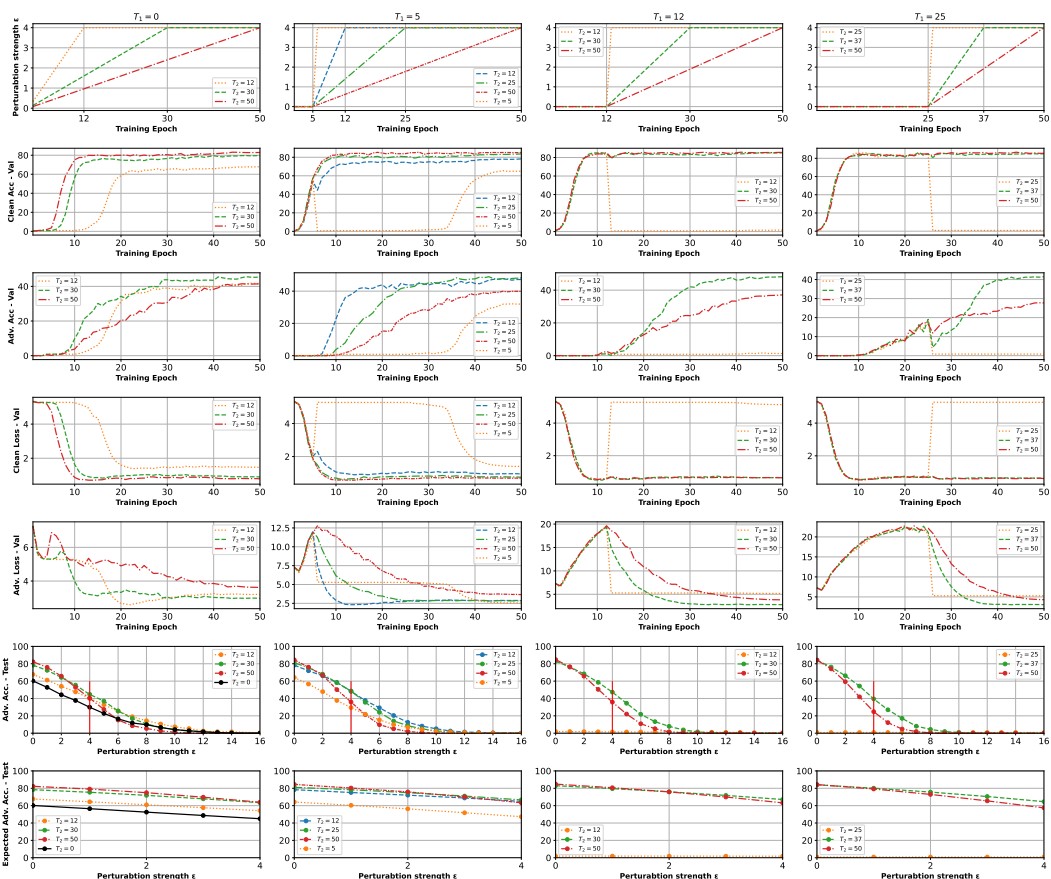

Figure 11: Effect of hyperparameters on `SWIN-Cars` for target $\varepsilon_g = {}^4/_{255}$. The numerical results are presented in Table 7. Same plot at $\epsilon_g = {}^8/_{255}$ are in Figure 12

| Model | Setting | $\epsilon = {}^4/_{255}$ | | | $\epsilon = {}^8/_{255}$ | | |
|---|---|---|---|---|---|---|---|
| | | Clean | Adv. | E. Adv. | Clean | Adv. | E. Adv. |
| ViT | fix | 37.29 | 16.89 | 26.49 | 15.59 | 5.84 | 10.12 |
| | scheduler | 70.96 | 22.66 | 46.42 | 63.91 | 11.63 | 34.38 |
| SWIN | fix | 56.40 | 32.08 | 44.35 | 35.64 | 14.21 | 24.16 |
| | scheduler | 79.78 | 39.50 | 60.90 | 72.55 | 22.87 | 47.49 |
| CNX | fix | 61.14 | 37.59 | 49.93 | 19.82 | 9.61 | 14.33 |
| | scheduler | 84.05 | 45.69 | 66.59 | 79.04 | 28.12 | 54.09 |
| R50 | fix | 37.26 | 16.99 | 26.62 | 22.88 | 8.82 | 15.09 |
| | scheduler | 67.76 | 20.13 | 42.65 | 55.57 | 10.48 | 29.84 |
| ClipViT | fix | 12.73 | 5.74 | 8.85 | 8.59 | 3.13 | 5.56 |
| | scheduler | 73.77 | 39.14 | 57.08 | 68.62 | 24.94 | 46.11 |
| ClipCNX | fix | 24.09 | 14.38 | 18.93 | 13.95 | 7.41 | 10.47 |
| | scheduler | 80.74 | 49.09 | 65.91 | 76.40 | 32.20 | 54.93 |

Table 10: Average per model of the clean accuracy, adversarial accuracy, and expected adversarial accuracy for $\epsilon = {}^4/_{255}$ and $\epsilon = {}^8/_{255}$.

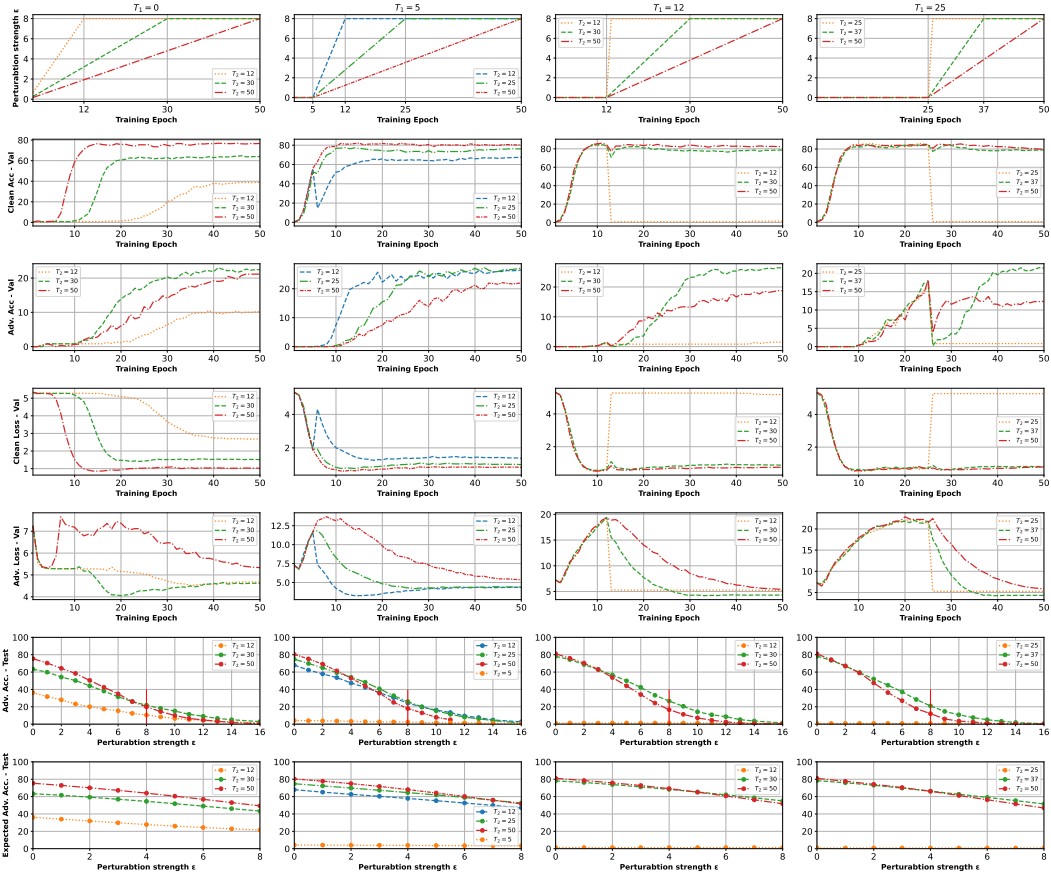

Figure 12: Effect of hyperparameters on SWIN-Cars for target $\varepsilon_g = 8/255$. The numerical results can be found in Table 7.

| Dataset | Setting | $\epsilon = 4$ | | | $\epsilon = 8$ | | |
|---|---|---|---|---|---|---|---|
| | | **Clean** | **Adv.** | **E. Adv.** | **Clean** | **Adv.** | **E. Adv.** |
| Aircraft | fix | 6.37 | 3.47 | 4.66 | 2.58 | 1.77 | 2.14 |
| | scheduler | 69.23 | 29.82 | 49.69 | 64.83 | 20.52 | 41.34 |
| Caltech | fix | 65.42 | 43.00 | 54.27 | 50.37 | 25.67 | 37.51 |
| | scheduler | 81.02 | 48.93 | 65.73 | 75.48 | 33.55 | 54.50 |
| Cars | fix | 25.73 | 14.22 | 19.99 | 4.65 | 2.50 | 3.45 |
| | scheduler | 82.43 | 45.33 | 65.29 | 77.25 | 28.45 | 54.16 |
| Cub | fix | 47.24 | 23.56 | 35.07 | 19.45 | 5.26 | 11.38 |
| | scheduler | 77.39 | 34.60 | 56.64 | 70.16 | 17.44 | 42.31 |
| Dogs | fix | 46.00 | 18.82 | 31.99 | 19.95 | 5.66 | 11.95 |
| | scheduler | 70.82 | 21.50 | 45.61 | 59.02 | 8.59 | 30.06 |

Table 11: Average per dataset of the clean accuracy, adversarial accuracy, and expected adversarial accuracy for $\epsilon = 4/255$ and $\epsilon = 8/255$.

| Model | Setting | $\epsilon = {}^4/_{255}$ | | | $\epsilon = {}^8/_{255}$ | | |
|-------|---------|-------|------|---------|-------|------|---------|
|       |         | **Clean** | **Adv.** | **E. Adv.** | **Clean** | **Adv.** | **E. Adv.** |
| SWIN  | fix       | 97.15 | 85.07 | 92.22 | 94.19 | 66.17 | 82.43 |
|       | scheduler | 98.62 | 85.48 | 93.67 | 97.50 | 69.08 | 86.93 |
| CNX   | fix       | 97.81 | 88.44 | 94.09 | 94.80 | 68.87 | 84.08 |
|       | scheduler | 99.29 | 88.23 | 95.36 | 98.27 | 71.32 | 89.03 |

Table 12: Imagenette results for $\epsilon = {}^4/_{255}$ and $\epsilon = {}^8/_{255}$.

