# OpenReview forum: "Robust Fine-Tuning from Non-Robust Pretrained Models: Mitigating Suboptimal Transfer With Epsilon-Scheduling"
_ICLR.cc/2026/Conference — ICLR 2026 Poster_

### Official Review · Reviewer_Vfsn · 2025-10-28

**Soundness:** 3
**Presentation:** 3
**Contribution:** 2
**Rating:** 4
**Confidence:** 4

**Summary:**

This paper explores finetuning a non-robust pretrained model with adversarial examples. The paper shows that direct adversarial finetuning, even with a small perturbation, leads to poor performance. To this end, the paper proposes epsilon-scheduling, which schedules perturbation strength as finetuning progresses. Experiments show that training curve under scheduling shows quicker clean accuracy convergence, achieves better tradeoff between accuracy and robustness, and leads to more optimal minima.

**Strengths:**

Robust finetuning of non-robust pretrained model is an important research question. The proposed method is simple, solid, and interesting. The method is well motivated from the observation of "delay time", a novel observation for robust finetuning. Experimental results show effective improvements for most cases. The visualization of training curve and converged minima are helpful for the understanding of the process.

**Weaknesses:**

1. The evaluation is limited to PGD adversarial training, which is known to be prone to overfitting and diminished clean accuracy in the finetuning setting. More advanced finetuning method like AutoLoRA [1] or CLAT [2] should be attempted to see if the proposed scheduling is also effective.
2. The current performance is highly relevant to baseline, task, and choice of hyperparameters. This makes the heuristic exploration of T1 T2 scheduling hard to be optimal for a new task and backbone. Some criteria to automatically decide when to start and end the epsilon scheduling based on things like gradient, loss, or accuracy would be helpful for better generalizability.
3. The evaluation is limited to finetuning on small-scale tasks, which may be an overkill with the large pretrained model. Having some exploration on ImageNet variants would better show the effectiveness of the proposed method under more complicated settings.

[1] Xu, X., Zhang, J., & Kankanhalli, M. (2023). Autolora: A parameter-free automated robust fine-tuning framework. arXiv preprint arXiv:2310.01818.

[2] Gopal, B., Yang, H., Zhang, J., Horton, M., & Chen, Y. Boosting Adversarial Robustness with CLAT: Criticality Leveraged Adversarial Training. In Forty-second International Conference on Machine Learning.

**Questions:**

1. Is there an automated way to decide when to start and end the scheduling based on the trianing progress?
2. How would the scheduling help with more recent robust finetuning methods and on larger tasks?

---

> ### Author Response · Authors · 2025-11-19
> **Response to Reviewer Vfsn (1/2)**
>
> Thank you for reviewing our paper. We are grateful you find our work deals with an important research question, is well motivated, with a simple method and solid results. Below are our responses to your concerns.
>
> **1.  PGD AT vs AutoLoRA**
>
> We would like to clarify that, to our knowledge, the observation regarding PGD adversarial training's susceptibility to overfitting pertains to classical adversarial training from scratch. The same phenomenon is not predominant in robust-finetuning. As a matter of fact, our extensive empirical analysis over multiple perturbation strengths show negligible overfitting for the non-robust backbones (See Figure 3).
>
> As for the suggested methods: AutoLoRA is specifically designed to preserve pretraining robustness in robust backbones with the help of a LoRA branch and CLAT is a dynamic layer selection (similar to RiFT, Zhu et al.) designed to operate on top of adversarial training. We note that these methods are specifically designed for robust backbones, however we do not claim that our Epsilon-Scheduling strategy improves fine-tuning on robust backbones. Instead, we claim that it is possible to achieve robust fine-tuning from non-robust backbones as exemplified by our Scheduling strategy. We believe that establishing this result is an important first step toward advancing downstream robustness in settings where the backbone is not robust, and we hope it will motivate the development of additional strategies tailored to this regime. We can make this message clearer in the paper and welcome suggestions on how to communicate it in the most effective way.
>
> **2.  Automated Scheduler**
>
> We discussed in section 6 (line 354 and 450) the effects of the value of $T_1$ and $T_2$. We recall that $T_1$ can be chosen sufficiently large to allow the model to converge on the downstream task performance. And $T_2$ must be chosen such that it starts low enough to prevent task adaptation, setting $T_2$ to final epoch is the safest choice. We considered that a validation free selection has merits for applicability in cases when validation adds computational burden without significant impact on the results. However, from the analysis, an automatic scheduler can be derived based on validation accuracy. A simple one can be described as follows:
>
> initialize with $\epsilon=0$ and then begin linear increase from $T_1$ to the end of training, where $T_1$ is automatically selected as the point at which the validation accuracy converges. Convergence is detected by monitoring the change in validation accuracy with a patience of 5 epochs and a tolerance of 2%.
>
> Below are some results for this automatic scheduler that we can add to our results. It effectively mitigates suboptimal transfer and yields overall good results.
>
> **Automated Scheduler**
>
> | Model | Setting | ε | Aircraft Clean | Aircraft Adv | Aircraft EAdv | Caltech Clean | Caltech Adv | Caltech EAdv | Cars Clean | Cars Adv | Cars EAdv |
> |-------|----------|----|----------------|--------------|----------------|----------------|--------------|----------------|-------------|-----------|------------|
> | SWIN | fix   | 4/255 | 7.70 | 4.80 | 6.11 | 79.97 | 57.16 | 69.19 | 60.20 | 29.70 | 44.74 |
> | SWIN | sched | 4/255 | 73.80 | 32.00 | 53.75 | 85.43 | 56.39 | 72.04 | 84.70 | 43.20 | 66.41 |
> | SWIN | auto  | 4/255 | 73.30 | 29.40 | 52.96 | 85.63 | 54.18 | 71.29 | 84.20 | 38.40 | 64.30 |
> | SWIN | fix   | 8/255 | 4.20 | 2.70 | 3.47 | 68.87 | 38.10 | 53.40 | 13.20 | 5.60 | 8.66 |
> | SWIN | sched | 8/255 | 69.20 | 22.40 | 45.12 | 80.27 | 38.67 | 60.26 | 78.00 | 23.50 | 53.57 |
> | SWIN | auto  | 8/255 | 68.40 | 18.60 | 42.69 | 81.71 | 35.82 | 59.92 | 79.20 | 18.70 | 51.43 |
> | CNX  | fix   | 4/255 | 7.60 | 4.50 | 5.86 | 83.27 | 61.54 | 73.08 | 69.60 | 43.20 | 57.52 |
> | CNX  | sched | 4/255 | 78.40 | 38.00 | 59.40 | 89.41 | 61.45 | 77.23 | 88.90 | 57.70 | 75.85 |
> | CNX  | auto  | 4/255 | 79.10 | 31.60 | 56.61 | 90.14 | 58.30 | 76.26 | 89.00 | 50.30 | 72.56 |
> | CNX  | fix   | 8/255 | 1.60 | 1.50 | 1.48 | 59.85 | 33.95 | 46.34 | 5.30 | 2.60 | 3.98 |
> | CNX  | sched | 8/255 | 75.00 | 28.80 | 50.90 | 84.99 | 41.82 | 64.92 | 85.60 | 35.90 | 65.04 |
> | CNX  | auto  | 8/255 | 76.20 | 23.60 | 49.65 | 86.48 | 38.39 | 64.35 | 86.20 | 29.90 | 63.49 |
>
> More generally, while the optimal choice of scheduling parameters ($T_1$, $T_2$) is indeed task- and backbone-dependent, we view this as a natural and impactful direction for future research rather than a limitation of our contribution. In fact, identifying adaptive or theoretically grounded criteria for when to start or end scheduling (e.g., via gradient statistics), discovering more effective parameterizations, or designing entirely new solutions to suboptimal transfer are all promising avenues for future work.

---

> > ### Author Response · Authors · 2025-11-19
> > **Response to Reviewer Vfsn (2/2)**
> >
> > **3.  Small Scale Task**
> >
> > If “small-scale” refers to the size of the datasets, this is precisely why they were selected. The low data regime reflects the constraints of limited data that occur in many domains and is a more realistic case of transfer learning. We have included the most challenging cases for robust fine tuning in literature, namely the fine-grained high resolution image classification tasks (eg: Stanford Cars, FGVC-Aircraft). A larger model is not necessarily an “overkill” for robust fine-tuning (cf. general comments).
> >
> > ImageNet variants are actually simpler settings for robust-finetuning since models often use ImageNet in the pretraining stage. Below are some results on Imagenette, that we propose to include in the appendix.
> >
> >
> > **Imagenette**
> >
> > | Model     | Setting    | Clean (ε=4/255) | Adv. (ε=4/255) | E.Adv. (ε=4/255) | Clean (ε=8/255) | Adv. (ε=8/255) | E.Adv. (ε=8/255) |
> > |-----------|------------|--------------|------------|---------------|--------------|------------|---------------|
> > | SWIN      | fix        | 97.15        | 85.07      | 92.22         | 94.19        | 66.17      | 82.43         |
> > | SWIN      | scheduler  | 98.62        | 85.48      | 93.67         | 97.50        | 69.08      | 86.93         |
> > | ConvNext  | fix        | 97.81        | 88.44      | 94.09         | 94.80        | 68.87      | 84.08         |
> > | ConvNext  | scheduler  | 99.29        | 88.23      | 95.36         | 98.27        | 71.32      | 89.03         |

---

> > > ### Comment · Reviewer_Vfsn · 2025-11-20
> > >
> > > I would like to thank the author for the responses and additional experiments. I appreciate the point made by the author about the importance of a method dedicated for finetuning a large clean pretrained model to a data-limited task. This builds the foundation for the significance of this work.
> > >
> > > With this understanding and given the quality of the paper, I would like to increase my score to 8 and advocate for acceptance.

---

### Official Review · Reviewer_SXJd · 2025-10-29

**Soundness:** 3
**Presentation:** 3
**Contribution:** 3
**Rating:** 6
**Confidence:** 5

**Summary:**

This paper studies robust fine-tuning (RFT) of non-robust pretrained models and observes suboptimal transfer, where robust objectives hinder task adaptation and degrade performance. The authors show that RFT initially delays learning, which results in the poor performance in clean performance. To address this, they propose Epsilon-Scheduling, a simple yet effective heuristic that gradually adjusts perturbation strength during training, improving both adaptation and robustness. Extensive experiments demonstrate consistent gains in expected robustness across multiple models and datasets.

**Strengths:**

The paper is well written and logically presented
The observation that RFT delays task adaptation is interesting, as it explains the low clean performance observed during RFT.
The proposed mitigation method is straightforward yet effective.

**Weaknesses:**

I noticed that in Figure 4, scheduler shows lower robustness compared to fixed, while in Table 1 it achieves higher adversarial robustness in most cases. Could the authors clarify this discrepancy or provide insights into why this happens?

**Questions:**

See weakness

---

> ### Author Response · Authors · 2025-11-19
> **Response to Reviewer SXJd**
>
> Thank you for reviewing our paper. We are grateful that you find our work well presented, interesting for RFT, with a simple yet effective method. Below is our response to your concern.
>
> **Figure 4 is a special case from Table 1.**
>
> Figure 4 highlights a typical case where there is an accuracy-robustness tradeoff (mainly R50-Caltech), i.e., the fix has better robust accuracy but lower clean accuracy at target epsilon (Table 1, 4/255, and Table 2, 8/255). The expected robustness metric comes as a complementary evaluation that shows the scheduler performs better on intermediate adversaries. This tradeoff situation occurs mainly in cases where suboptimal transfer is mild, most often ResNet50 (potentially higher inductive bias) and Caltech (easy task, diverse object recognition). In fact, in the majority of cases, the scheduler clearly improves robust accuracy too.

---

> > ### Comment · Reviewer_SXJd · 2025-11-19
> >
> > Thanks for the response. It makes sense to me and I raised my score to 8.

---

### Official Review · Reviewer_AYiL · 2025-11-03

**Soundness:** 2
**Presentation:** 3
**Contribution:** 2
**Rating:** 4
**Confidence:** 4

**Summary:**

This paper studies how to improve adversarial robustness via robust fine-tuning when the pre-trained model is vanilla/non-robust. Firstly, the authors identify the phenomenon, “suboptimal transfer”, when standard fine-tuning with adversarial training leads to low clean accuracy on downstream tasks. Since this can be attributed to the delayed task adaption by the authors when robust objectives are applied from the start of the fine-tuning, they propose Epsilon-Scheduling, a two-hinge linear schedule that starts with standard fine-tuning before gradually increasing perturbation strength to the target epsilon. Finally, the authors introduce expected robustness, a metric that evaluates model accuracy across the full perturbation range. Epsilon-Scheduling consistently shows improved expected robustness compared to fixed-perturbation RFT with the experimental results.

**Strengths:**

- **Studying an important and practical problem.** The focus of the paper on non-robust pretrained models is highly relevant. Nearly all publicly available pretrained foundation models are not adversarially trained. Improving robust fine-tuning techniques with those models would help model security in practice.
- **Interesting insights on task adaptation delay and proposing simple yet effective solution.** The observation that robust fine-tuning delays task adaption (figure 3) is well-introduced. Especially, the strong correlation (>90%) between delay time and suboptimal transfer severity provides valuable understanding of the underlying mechanisms. This makes the idea of the proposed epsilon-scheduling method intuitive. It is an easy approach which generalizes prior linear warmup strategies.
- **New evaluation metric, strong reproducibility.** Expected robustness helps measure the accuracy-robustness tradeoff than point estimates at a certain epsilon perturbation alone. It is well-grounded in a realistic threat model. Also, an anonymized code repository is provided with detailed hyperparameters and configurations to support reproducibility for future research in the field.

**Weaknesses:**

We thank the authors for submitting the paper to ICLR 2026! There are a few weaknesses listed below which I believe can make the paper better.
- **Limited theoretical understanding.** The paper is purely based on empirical results. While those observations are compelling, the paper lacks theoretical analysis of why robust objectives delay task adaptation or why Epsilon-Scheduling finds better local minima. Some theoretical perspective can make the claim (such as figure 7) much more grounded.
- **Lack of comparisons with other robust fine-tuning strategies.** There’s no comparison with other methods mentioned in the related works section such as TWINS, AutoLoRA, or RoLi. It would be interesting to see how it competes with other RFT methods and if they could be combined to make further improvements given the proposed scheduling method could be compatible with other architectural changes.
- **Concerns on the expected robustness metric.** Since it is aiming to measure the trade-off between robustness and accuracy, it would be good to consider cases with very high clean accuracy, when the metric could be overemphasizing clean accuracy. Also, some discussion on how to make decisions when / whether to use this metric vs. traditional worse-case robustness would be informative for practitioners.

**Questions:**

- Could you provide some theoretical intuition/hypothesis on why robust objectives delay adaptation would occur? For example, is there a connection to the loss landscape geometry? Are there other factors that could be involved – size of the dataset, optimization objectives, etc?
- Can you provide direct comparisons with TWINS, AutoLoRA, or RoLi with the same experimental setup? Even if they assume robust backbones, understanding relative performance would be valuable.
- Can you add some error bars/confidence intervals with multiple runs? Since sometimes the values are very close to each other, this will help mitigate noises.

---

> ### Author Response · Authors · 2025-11-19
> **Response to Reviewer AYiL (1/3)**
>
> Thank you for reviewing our paper. We are grateful that you find our work highly relevant, with interesting insights and strongly reproducible. Below are our responses to your concerns.
>
> **1.  Theoretical Understanding**
>
> This work formalized the suboptimal transfer in robust fine-tuning from non-robust pretrained models and pinpointed the delayed task adaptation as a possible cause. In Section 6.2 and Figure 6, we have provided an explanation of this phenomenon based on the discrepancy between the two loss landscapes for the clean loss and the robust loss in the vicinity of the pretrained model. This explanation aligns with the loss curves and is also consistent with “divergent gradients” in robust fine-tuning from Xu et al. 2023.
>
> We concur that further research is essential to enhance our understanding of the learning dynamics of robust fine-tuning, and that establishing connections with other phenomena would represent a valuable contribution. However, given the limited theoretical understanding of the success of robust fine-tuning from robustly pretrained models, we believe that even our empirical findings alone represent an important contribution towards a better understanding of robustness in this context. Our findings open a number of new important open questions, which we propose to articulate more explicitly in the manuscript:
>
> **(i)** What mechanisms underlie suboptimal transfer: is delayed task adaptation the only cause of suboptimal transfer or are there other factors?
>
> **(ii)** Can we find other approaches to mitigate delayed task adaptation different from Epsilon-Scheduling?
>
> **(iii)** What mathematical theory can account for suboptimal transfer or delayed task adaptation?
>
> **(iv)** If robust pretraining is not indispensable, what specific properties (if any) in pretraining really matter for downstream robustness and allow effective robust fine-tuning?
>
> We believe that these questions represent promising avenues enabled and motivated by our work, and we expect an increased attention for robust fine-tuning from non-robust backbones to lead to even more effective strategies.
>
> **2.  Comparison with Twins, AutoLoRa, Roli**
>
> As you noted, all of the above strategies explicitly rely on robust pretraining and are unsuitable for non-robust backbones, as acknowledged by the authors themselves:
>
> -   TWINS (Liu et al. 2023) decouples the batch statistics from pretraining data and current task to preserve robustness from large-scale robust pretraining. Liu et al. do not apply it to non-robust backbones claiming that “robust pre-training is indispensable to downstream robustness”.
>
> -   AutoLoRA (Xu et al. 2023) disentangles the robust pretrained features from current task features using a LoRA branch. The authors show that robust pretrainning is necessary and the worst performance happens for the non-robust pretraining (epsilon_pretraining=0, Xu et al. Table 4).
>
> -   RoLi (Hua et al. 2024) initializes the head with a robustly fine-tuned linear head. The authors explicitly argue that robust pretraining is a prerequisite by showing that linear probing with a robust objective on a non-robust backbone fails dramatically and therefore does not provide a good initialization (Hua et al., 2024, Figure 3).
>
>
> For robust backbones, it is important to note that we do not claim that Epsilon-Scheduling is a general strategy that also works for them. Our analysis in Section 6.2 (line 431 & Table 3) shows that robust backbones do not suffer as much suboptimal transfer as their non-robust counterparts. However, below are some elements of comparison for robust backbones. These results simply confirm our analysis in section 6.2 (line 431 & Table 3) that, on robust backbones, the scheduler, as proposed, often improves accuracy at the expense of robustness compared to RFT-fix. The question of whether there exists a scheduling strategy that can match or outperform RFT-Fix and the above-mentioned strategies for robust backbones is an interesting avenue for future research.
>
>  **Robust ResNet50 × Caltech (Xu et al. 2023)**
>
> | Method            | Clean Acc | Robust Acc           |
> |-------------------|-----------|-----------------------|
> | Fix (Xu et al.)   | 66.78     | 47.76 (PGD-10)        |
> | TWINS             | 69.41     | 47.79 (PGD-10)        |
> | AutoLoRA          | 69.85     | 47.82 (PGD-10)        |
> | Fix (ours)        | 65.25     | 36.60 (APGD-10)      |
> | Scheduler         | 71.90     | 34.08 (APGD-10)       |
>
> **Robust SWIN × Caltech (Hua et al. 2024)**
>
> | Method           | Clean Acc | Robust Acc          |
> |------------------|-----------|----------------------|
> | Fix Hua et al.   | 77.35     | 49.13 (PGD-10)       |
> | RoLi             | 83.78     | 55.42 (PGD-10)       |
> | Fix (ours)       | 77.39     | 45.87 (APGD-10)      |
> | Scheduler        | 81.86     | 43.43 (APGD-10)      |

---

> > ### Author Response · Authors · 2025-11-19
> > **Response to Reviewer AYiL (2/3)**
> >
> > **3.  Concern about the Expected Robustness Metric**
> >
> > High clean accuracy does not necessarily bias the expected robustness; for example, a standardly fine-tuned model has high clean accuracy but close to zero robustness on even small adversaries, which leads to a very small area under the accuracy curve.
> >
> > The expected robustness comes as a complementary evaluation when there is an accuracy-robustness tradeoff, rather than a replacement of the traditional worst-case robustness.
> >
> > We recall that the value of the expected robustness has a distinct interpretation. It summarizes robustness over all adversarial strengths up to the target \epsilon and, by definition, corresponds to a threat model where inputs are subjected to attacks with a variable perturbation strength (including zero perturbation). In our current formulation, each adversarial strength is weighted equally through a uniform distribution over the range. As noted in line 322, practitioners may instead choose a different probability distribution to emphasize accuracy or robustness. For example, with a Dirac distribution centered at 0 (epsilon-target), it falls back to clean accuracy (worst case robust accuracy).
> >
> > **4.  Statistical Significance**
> >
> > We believe the diversity of experimental conditions provides compelling evidence of the efficacy of epsilon scheduling in mitigating suboptimal transfer and yielding good results. While there are cases with similar robust accuracies, these are fully aligned with our message and do not affect the main claims of our work. In fact, at a high perturbation regime, almost all ambiguity disappears, and the scheduler becomes the only viable solution. For the moderate perturbation regime, we recall that for robust accuracy, the scheduler is better in 21 out of 30 (70%) and consistently better in clean accuracy and expected robustness. We provide below the averages per model and per dataset, and the t-test statistics for the results. We will add a section in the appendix on statistical significance and update the results description.
> >
> > **T-Statistics**
> >
> > | Metric     | n_pairs | t-stat (ε=4) | p-value (ε=4)        | t-stat (ε=8) | p-value (ε=8)        |
> > |------------|---------|--------------|------------------------|--------------|------------------------|
> > | Clean Acc  | 30      | 7.823294     | 1.255470e-08          | 12.387491    | 4.170254e-13          |
> > | Adv.       | 30      | 4.348780     | 1.540867e-04          | 5.447550     | 7.317049e-06          |
> > | E. Adv.    | 30      | 6.595568     | 3.153155e-07          | 9.270810     | 3.572919e-10          |
> >
> > **Averages per model**
> >
> > | Model          | Setting    | Clean (ε=4) | Adv. (ε=4) | E.Adv. (ε=4) | Clean (ε=8) | Adv. (ε=8) | E.Adv. (ε=8) |
> > |----------------|------------|--------------|------------|---------------|--------------|------------|---------------|
> > | VIT            | fix        | 37.292       | 16.886     | 26.494        | 15.518       | 5.840      | 10.116        |
> > | VIT            | scheduler  | 70.964       | 22.662     | 46.416        | 63.912       | 11.634     | 34.384        |
> > | SWIN           | fix        | 56.402       | 32.084     | 44.352        | 35.642       | 14.216     | 24.156        |
> > | SWIN           | scheduler  | 79.784       | 39.504     | 60.904        | 72.552       | 22.874     | 47.486        |
> > | ConvNext       | fix        | 61.142       | 37.586     | 49.932        | 19.820       | 9.612      | 14.328        |
> > | ConvNext       | scheduler  | 84.054       | 45.690     | 66.594        | 79.044       | 28.116     | 54.088        |
> > | R50            | fix        | 37.262       | 16.994     | 26.620        | 22.884       | 8.820      | 15.092        |
> > | R50            | scheduler  | 67.764       | 20.134     | 42.648        | 55.568       | 10.482     | 29.836        |
> > | ClipViT       | fix        | 12.730       | 5.744      | 8.852         | 8.586        | 3.134      | 5.560         |
> > | ClipViT       | scheduler  | 73.770       | 39.136     | 57.076        | 68.620       | 24.938     | 46.110        |
> > | ClipCNX  | fix        | 24.090       | 14.378     | 18.930        | 13.954       | 7.408      | 10.468        |
> > | ClipCNX  | scheduler  | 80.740       | 49.088     | 65.912        | 76.396       | 32.202     | 54.936        |

---

> > > ### Author Response · Authors · 2025-11-19
> > > **Response to  Reviewer AYiL (3/3)**
> > >
> > > **Averages per dataset**
> > >
> > > | Dataset   | Setting    | Clean (ε=4) | Adv. (ε=4) | E.Adv. (ε=4) | Clean (ε=8) | Adv. (ε=8) | E.Adv. (ε=8) |
> > > |-----------|------------|--------------|------------|---------------|--------------|------------|---------------|
> > > | Aircraft  | fix        | 6.37         | 3.47       | 4.66          | 2.58         | 1.77       | 2.14          |
> > > | Aircraft  | scheduler  | 69.23        | 29.82      | 49.69         | 64.83        | 20.52      | 41.34         |
> > > | Caltech   | fix        | 65.42        | 43.00      | 54.27         | 50.37        | 25.67      | 37.51         |
> > > | Caltech   | scheduler  | 81.02        | 48.93      | 65.73         | 75.48        | 33.55      | 54.50         |
> > > | Cars      | fix        | 25.73        | 14.22      | 19.99         | 4.65         | 2.50       | 3.45          |
> > > | Cars      | scheduler  | 82.43        | 45.33      | 65.29         | 77.25        | 28.45      | 54.16         |
> > > | Cub       | fix        | 47.24        | 23.56      | 35.07         | 19.45        | 5.26       | 11.38         |
> > > | Cub       | scheduler  | 77.39        | 34.60      | 56.64         | 70.16        | 17.44      | 42.31         |
> > > | Dogs      | fix        | 46.00        | 18.82      | 31.99         | 19.95        | 5.66       | 11.95         |
> > > | Dogs      | scheduler  | 70.82        | 21.50      | 45.61         | 59.02        | 8.59       | 30.06         |
> > >
> > > Note that it is standard practice in adversarial robustness evaluation to report results without error bars. In our setting, running multiple seeds for more than 80 configurations would be computationally prohibitive. If any error bars are desired, we would appreciate guidance on which configurations to prioritize.
> > >
> > > *References*:
> > > *Liu, Ziquan, et al. "Twins: A fine-tuning framework for improved transferability of adversarial robustness and generalization." CVPR 2023.*
> > >
> > > *Xu, Xilie, Jingfeng Zhang, and Mohan Kankanhalli. "Autolora: A parameter-free automated robust fine-tuning framework." arXiv preprint arXiv:2310.01818 (2023).*
> > >
> > > *Hua, Andong, et al. "Initialization matters for adversarial transfer learning." CVPR 2024.*

---

### Author Response · Authors · 2025-11-19
**General comment to all reviewers**

We thank you for reviewing our paper and providing constructive comments, which are helpful to improve the clarity and rigor of our work.

We are grateful that you recognize the high relevance of our work, namely, that robust fine-tuning of non-robust pretrained models is an important research question. As you have also noticed, to the best of our knowledge, previous research on robust fine-tuning strategies assumes that starting from a robustly pretrained model is necessary. By questioning this prevalent assumption, our paper aims to stimulate additional research in this critical area. While robustness inheritance is effective, we show that it is not a prerequisite. Robust fine-tuning is achievable even when starting from a non-robust pretrained model. We support our claim with rigorous analysis and compelling empirical evidence to pave the way for further research in this direction and to open doors to even more effective strategies.

Based on your feedback, we would also like to highlight three observations that provide more context for our results:

-   **High Clean Accuracy from Non-Robust Backbones**: To the best of our knowledge, we demonstrate, for the first time, that robust fine-tuning of a non-robust backbone can achieve clean accuracy on par with, or even surpassing, models derived from robust backbones.

-   **Challenging the Robust-Small vs. Non-Robust-Large Paradigm**: We provide evidence against earlier findings that suggested that *“opting for a smaller yet robust pretrained model proves to be a more effective strategy for attaining adversarial robustness in downstream tasks compared to using a larger, non-robust pretrained model”* (Hua et al. 2024). The CLIP-ConvNeXt on Cars at both 4/255 and 8/255 (Tables 1 & 2) provides such evidence: while RFT-Fix fails dramatically, our simple scheduling strategy surpasses all robust backbones in both clean and robust accuracy (Table 3).

-   **Cross-Architectural Empirical Evaluation**: We demonstrate the effectiveness of our robust fine-tuning strategy across both convolutional and transformer architectures. To our knowledge, this is the first study of a robust fine-tuning strategy that makes claims applicable across multiple architecture types.

We believe that our contribution can be crucial in advancing practical robustness in downstream tasks. We address individual concerns below.

Reference:
Hua, Andong, et al. "Initialization matters for adversarial transfer learning." CVPR 2024.

---

### Author Response · Authors · 2025-12-03
**New version**

We would like to sincerely thank reviewers **SXJd** and **Vfsn** for raising their scores to 8 and for actively advocating for acceptance. In response to the reviewers’ feedback, we have updated the paper with additional results, analyses, and clarifications. Below, we summarize the changes in order of appearance in the manuscript:
### Section 2: Related Work
- Added three sentences to clarify the limitations and scope of the previous strategies.
### Section 5: Paragraph on Expected Robustness Evaluation
- Added one sentence clarifying the choice of distribution
### Section 6.3 – Ablation and Sensitivity Analysis
- Added a new paragraph on the **Automated Scheduler**, including results from the automated schedule derived from our theoretical analysis.
- **Table 4**: New results for the **AutoScheduler** on **SWIN** and **ConvNext** across **Caltech**, **Aircraft**, and **Cars** at **4/255** and **8/255**.

### Conclusion – Limitations & Future Directions
- Added a new final paragraph discussing the broader open questions highlighted by our findings and outlining promising directions for future work.

### Appendix D – Statistical Significance (New)
- **Table 9**: Added **t-test statistics** and **p-values** comparing **RFT-Fix** and **RFT-Scheduler**.
- **Table 10**: Added **average results per model**.
- **Table 11**: Added **average results per dataset**.

### Appendix E – Additional Results on ImageNette (New)
- **Table 12**: Added results for **all models** on ImageNette at **4/255** and **8/255**.

---

### Meta-Review · Area_Chair_G6XH · 2026-01-06

**Summary:**

The paper investigates robust fine-tuning (RFT) of non-robust pretrained models, identifying a ''suboptimal transfer'' phenomenon where standard adversarial training objectives hinder task adaptation. To mitigate this, the authors propose Epsilon-Scheduling, a linear schedule that gradually increases perturbation strength. They also introduce Expected Robustness as a more comprehensive evaluation metric.

The reviewers were impressed by the identification of the ''suboptimal transfer'' phenomenon and the simplicity and effectiveness of the proposed Epsilon-Scheduling method. The addition of the Expected Robustness metric was also noted as a valuable contribution to the field of robust fine-tuning.

**Reviewer Concerns:**

The authors provided a very thorough rebuttal that successfully addressed most technical concerns.

Still outstanding concerns:
While the paper remains primarily empirical, the authors acknowledged the limited theoretical understanding of the loss landscape geometry as an avenue for future work. This is acceptable given the novelty of the empirical observations.

**Reviewer Scores:**

1 Both Reviewer SXJd and Vfsn raised the score after authors' responses.

2 Although Reviewer AYIL did not explicitly update their score in the provided text, the authors addressed the main concerns regarding baselines and statistical significance with extensive new data.

---

### Decision · Program_Chairs · 2026-01-26

Accept (Poster)